# Sequence-agnostic Continual Multi-modal Clustering

## Abstract

Continual multi-modal clustering (CMC) aims to address the challenges posed by the continuous arrival of multi-modal data streams, enabling models to progressively update cluster assignments while avoiding catastrophic forgetting. CMC closely aligns with the requirements of real-world scenarios and has attracted significant attention from researchers. However, existing CMC methods face two limitations. (1) They fail to reliably model the relationship between historical and new information, leading to redundancy in the shared representation and weakened discriminative power of clustering. (2) They are highly sensitive to modality sequence, as early high-quality modalities are gradually forgotten, making the results dependent on the input order. To address these limitations, we propose a novel Sequence-agnostic Continual Multi-modal Clustering (SCMC) method that achieves reliable continual learning and is insensitive to the modality arrival sequence. Specifically, SCMC employs a residual fusion network to suppress the update bias introduced by the newly arrived modalities. It then leverages a cross-temporal knowledge collaboration mechanism to bidirectionally filter information between the historical information and the new modalities, thereby maximizing the preservation of task-relevant information and ensuring reliable continual learning. To eliminate the high sequence sensitivity, we design a sequence-agnostic anti-forgetting strategy, which aligns the current features and cluster distribution with the previous step through cross-temporal consistency transfer, and then prioritizes retaining high-value modality information based on modality importance scores. Extensive experiments demonstrate that SCMC outperforms existing SOTA methods, exhibiting sequence insensitivity and strong anti-forgetting capabilities. To the best of our knowledge, SCMC is the first approach to explicitly address the sequence sensitivity problem in CMC.

## 1 Introduction

Continual multi-modal clustering (CMC) has emerged as a crucial paradigm to address the growing need for learning from ever-expanding and streaming multi-modal data. Unlike traditional static clustering methods that fail to adapt to dynamic data flows and the gradual expansion of modalities, CMC simultaneously captures cross-modal complementary semantics and preserves accumulated knowledge in the face of dynamic flow data, which is more closely integrated with real-world application scenarios. Due to these advantages, CMC has shown promising utility in various domains such as medical diagnosis (Ye et al., 2024), visual question answering (Qian et al., 2023), and 3D semantic segmentation (Cao et al., 2023).

Catastrophic forgetting remains a fundamental challenge in continual learning, and existing CMC methods have explored various strategies to address it. Wan et al. (2022) proposed maintaining a consensus partition matrix that is iteratively updated as new views arrive, thereby preserving cluster consistency and alleviating forgetting induced by new modalities. Li et al. (2024b) employed self-distillation to sustain prototype-sample relationships, ensuring stable knowledge retention when learning additional tasks. Wan et al. (2024) introduced a data buffer to store filtered structural information, and then generated a robust partitioning matrix through contrastive learning to enhance the consistency and complementarity of multi-modal information. Zhang et al. (2025a) combined low-dimensional representation learning with consistency of cluster assignments, facilitating exploration of incremental views while strengthening consistent knowledge retention. Cha et al. (2021) integrated contrastive learning with self-supervised distillation to jointly learn representations and

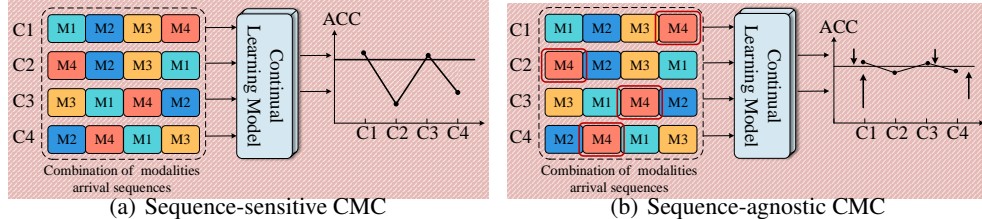

(a) Sequence-sensitive CMC        (b) Sequence-agnostic CMC

Figure 1: The motivation of SCMC, where $\{Mi\}_i^{i=4}$ represents the different modalities that continuously arrive, $\{Ci\}_i^{i=4}$ represents the arrival sequence of different modalities, and M4 represents the most informative modality. (a) shows that the arrival sequences of different modalities can produce significantly different clustering results. (b) shows that SCMC is stable in clustering different modality sequences.

preserve prior knowledge, thus preventing newly arrived modalities from destabilizing previously acquired representations. Cai et al. (2023) designed a view-specific knowledge base that captures intra-view knowledge, effectively avoiding excessive storage while ensuring continuous knowledge inheritance.

However, existing CMC methods have two limitations. **First, they lack reliability in integrating historical knowledge with newly arrived information.** Many approaches adopt straightforward concatenation or fixed-weight linear fusion, which implicitly assumes modality homogeneity and overlooks the impact of noise. Such simplistic strategies neither assess the uncertainty of new modalities nor suppress redundant or conflicting signals. Consequently, the shared representation is gradually contaminated with noise and task-irrelevant factors, leading to diminished clustering discriminability as learning proceeds. Developing a reliable continual learning framework that effectively preserves informative content while adaptively filtering redundancy thus remains a critical challenge. **Second, current methods exhibit strong sensitivity to the sequence of modality arrival, particularly when high-quality modalities arrive earlier than low-quality ones.** By treating all modalities equally during knowledge distillation, low-quality signals are inevitably preserved alongside valuable information, amplifying the risk of catastrophic forgetting. The motivation of SCMC is shown in Figure 1, which is more in line with real-world scenarios. Therefore, it is essential to design unforgettable high-quality modalities to ensure that discriminative knowledge is always preserved throughout the learning process.

To address the above limitations, we propose a novel Sequence-agnostic Continual Multi-modal Clustering (SCMC) method, designed to enable reliable continual learning while mitigating sensitivity to modality sequence. We first design a residual fusion network that mitigates the update bias introduced by newly arrived modalities. Through residual channels, the network preserves previously learned discriminative subspaces and establishes a high-rank global basis capable of accommodating new information.

Building on this, we introduce a cross-temporal knowledge collaboration mechanism that performs bidirectional information filtering between the historical fused representation and the high-level features of new arrival modalities. This mechanism enhances complementary signals while suppressing redundancy, thereby reducing temporal noise accumulation and ensuring balanced utilization of discriminative features. To further prevent the loss of early high-quality modalities, we develop a sequence-agnostic anti-forgetting strategy comprising two components: (1) cross-temporal consistency transfer, which aligns current representations and cluster assignments with their historical counterparts to ensure smooth knowledge inheritance and stable decision boundaries; and (2) quality-aware historical consolidation, which employs modality importance scores to selectively retain and update historical information, prioritizing high-quality modalities while mitigating interference from low-quality ones. In summary, SCMC achieves a reliable integration of historical and current information, explicitly eliminates sequence bias, and provides stable and sequence-invariant clustering performance in an unsupervised manner. The main contributions of this work are as follows:

- Unlike existing CMC methods that remain sensitive to the sequence of modality arrival, this paper introduces a novel SCMC framework that successfully enables reliable contin-

ual learning in streaming settings without supervision, explicitly addressing the sequence-agnostic problem.

- A new cross-temporal knowledge collaboration mechanism is designed to align historical fusion representations with new modal features bidirectionally. By filtering out redundancy, this mechanism effectively suppresses the accumulation of temporal noise while enhancing the balance between feature utilization and discriminability.

- A sequence-agnostic anti-forgetting strategy is proposed that combines cross-temporal consistency transfer with quality-aware integration. By weighting modal importance, the strategy can preserve high-value signals, mitigate forgetting, and achieve sequence-invariant clustering.

- To the best of our knowledge, this paper is the first to explicitly tackle the sequence-agnostic challenge in continual multi-modal clustering, opening new directions for robust unsupervised learning under dynamic multi-modal streams.

## 2 RELATED WORK

**Deep Multi-modal Clustering.** Deep multi-modal clustering (DMC) leverages complementary information across heterogeneous modalities for unsupervised grouping. Unlike unimodal clustering, DMC integrates multi-source information to enable more comprehensive and robust knowledge discovery. Existing methods include: matrix factorization-based approaches (Zhang et al., 2025c; Yang et al., 2024; Wen et al., 2024a) that decompose feature matrices for latent alignment, shared representation-based approaches (Huang et al., 2019; Pan & Kang, 2023; Chen et al., 2025) that employ autoencoders or contrastive learning for unified semantic spaces, and graph learning-based approaches (Wang et al., 2022b; Wen et al., 2024b; Zhang et al., 2025b) that model structural dependencies via cross-modal graphs and GNNs. Despite progress, effectively exploiting cross-modal complementarity while suppressing redundancy and conflict remains a core challenge.

**Continual Learning.** Continual learning (CL) enables efficient knowledge transfer and sharing under dynamic data streams. Main approaches include: regularization-based methods (Jung et al., 2020; Zhou et al., 2024; Lewandowski et al., 2025) that constrain parameters to preserve past knowledge, memory replay-based methods (Wang et al., 2022a; Graffieti et al., 2023; Wang et al., 2025b) that approximate historical distributions, and knowledge distillation-based methods (Phan et al., 2022; Li et al., 2024a; Yang et al., 2025) that balance old and new knowledge via distillation loss. Although these methods have alleviated the forgetting problem to a certain extent, catastrophic forgetting remains a core challenge that needs to be addressed in continual learning, especially in scenarios with long task sequences or highly complex modalities.

Nowadays, researchers have begun to introduce the idea of continual learning into multi-modal clustering to form continual multi-modal clustering. However, existing methods still suffer from two shortcomings: First, they lack reliable modeling of historical and new information, leading to the accumulation of redundancy and weakening clustering discriminability. Second, they are highly sensitive to input sequence, making high-quality modalities easily forgotten in subsequent learning. To address this, we propose a novel sequence-agnostic continual multi-modal clustering method that successfully achieves reliable knowledge preservation and sequence insensitivity in streaming scenarios.

## 3 METHODOLOGY

**Problem Formulation.** Given multi-modal inputs $\mathcal{X} = \{\mathbf{X}^1, \ldots, \mathbf{X}^T\}$, each modality is pre-trained with an encoder-decoder to obtain latent $\mathbf{Z}^t$, mapped to features $\mathbf{H}^t = f_\theta(\mathbf{Z}^t)$ and clustered as $\mathbf{C}^t = g(\mathbf{H}^t)$. Historical features are aggregated via a residual fusion network into $\mathbf{H}^f_{\text{old}}$, which interacts with new features through cross-temporal knowledge collaboration to form $\mathbf{H}^f$. Quality-aware consolidation preserves reliable historical signals, and $\mathbf{H}^f$ is clustered to yield global assignments $\mathbf{Q} = g(\mathbf{H}^f)$, ensuring sequences-agnostic clustering. The main notations of SCMC are shown in Table 7.

**Overall Framework.** Figure 2 shows the SCMC framework. When a new modality arrives, SCMC integrates current and historical representations via a reliable continual information propagation

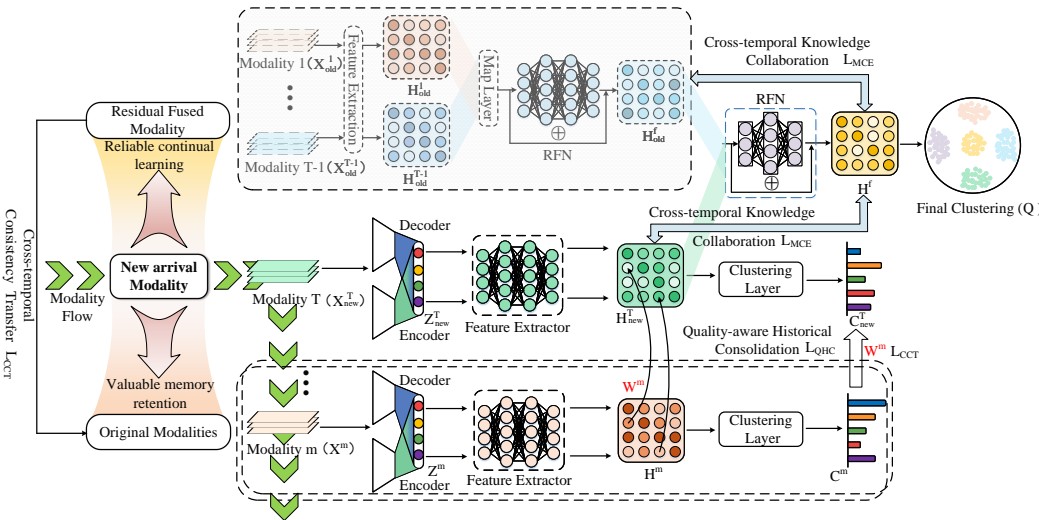

Figure 2: Overall framework of the proposed SCMC. A new modality $\mathbf{X}_{\text{new}}^{t}$ is first pre-trained with an autoencoder to obtain $\mathbf{Z}_{\text{new}}^{t}$ and then mapped to high-level features $\mathbf{H}_{\text{new}}^{t}$. To enhance cross-temporal collaboration, we maximize the matrix mutual information between $\mathbf{H}^{f}$, and $\mathbf{H}_{\text{old}}^{f}$, $\mathbf{H}_{\text{new}}^{t}$, ensuring bidirectional alignment and redundancy suppression. To mitigate forgetting of early high-quality modalities, we introduce a sequence-agnostic anti-forgetting strategy with (i) cross-temporal consistency transfer, aligning current features and assignments with prior steps, and (ii) quality-aware consolidation, selectively retaining historical information via modality-importance weights $W^{m}$. The refined $\mathbf{H}^{f}$ is finally passed to the clustering layer for stable, sequence-agnostic cluster assignments $\mathbf{Q}$.

module (Sec. 3.1) and applies a sequence-agnostic anti-forgetting strategy (Sec. 3.2) to retain high-value knowledge while suppressing low-quality interference. The model is optimized with the Adam optimizer, producing stable and sequence-agnostic clustering performance (Sec. 3.3).

## 3.1 RELIABLE CONTINUAL INFORMATION PROPAGATION FRAMEWORK

Existing CMC methods often lack explicit mechanisms to ensure reliability when fusing historical and newly arrived information, which results in noise accumulation and degradation of discriminability in the shared representation. The reliable continual information propagation (RCIP) framework addresses this by explicitly modeling reliability to enable dependable knowledge transfer and preserve discriminative power in streaming scenarios. RCIP consists of two parts: (i) a residual fusion network (RFN) that preserves discriminative subspaces while incorporating new modalities into a stable global basis to reduce update bias, and (ii) a cross-temporal knowledge collaboration (CTKC) that filters information bidirectionally between historical and current features, enhancing complementarity and suppressing redundancy in the shared space.

**Residual Fusion Network (RFN).** RFN retains the learned discriminative subspace through the residual path to alleviate the update bias of the newly arrived modality. In practice, RFN operates in two stages. First, multiple historical modality features are fused into a stable global basis:

$$\mathbf{H}_{\text{old}}^{f} \;=\; \mathcal{F}\big(\mathbf{H}_{\text{old}}^{1},\, \mathbf{H}_{\text{old}}^{2},\, \ldots,\, \mathbf{H}_{\text{old}}^{t-1}\big). \tag{1}$$

Second, the historical basis and the newly arrived modality are residually fused:

$$\mathbf{H}^{f} \;=\; \mathcal{F}\big(\mathbf{H}_{\text{old}}^{f},\, \mathbf{H}^{t}\big). \tag{2}$$

Let the concatenated multi-modal representation be $\mathbf{M} \in \mathbb{R}^{d \times B}$ with $d$ feature dimensions and batch size $B$ (samples as columns). RFN adopts a bottleneck down-up residual block $\mathcal{F}$ is:

$$\mathcal{F}(\mathbf{M}) \;=\; \mathbf{M} \;+\; \sigma\Big(\text{Up}\big(\sigma\big(\text{Down}([\mathbf{M}])\big)\big)\Big), \tag{3}$$

where $\sigma = (\cdot, \cdot)$ is the activation function, ReLU is selected. Down $: \mathbb{R}^{d \times B} \to \mathbb{R}^{d/2 \times B}$, Up $: \mathbb{R}^{d/2 \times B} \to \mathbb{R}^{d \times B}$. The identity term $\mathbf{M}$ explicitly preserves the high-rank fused basis constructed from historical modalities, while the non-linear learns low-rank corrections driven by the current modality. This design allows new modalities to refine but not overwrite previously learned discriminative directions, which is crucial for reliable information preservation in continual fusion.

**Cross-Temporal Knowledge Collaboration.** In continual multi-modal learning, the historical fused representation and the high-level features of a newly arrived modality are simultaneously complementary and potentially conflicting. Preserving complementarity while suppressing redundancy is essential for reliable shared representations. CTKC performs bidirectional information filtering along two cross-temporal channels (fusion $\leftrightarrow$ history and fusion $\leftrightarrow$ current), guiding the shared representation $\mathbf{H}^f$ to absorb complementary signals from both sources while suppressing redundant components that would otherwise contaminate the shared space, thereby enhancing the reliability of continual learning.

Concretely, CTKC forms two channels $(\mathbf{H}^f, \mathbf{H}^f_{\text{old}})$ and $(\mathbf{H}^f, \mathbf{H}^t_{\text{new}})$, where $\mathbf{H}^f, \mathbf{H}^f_{\text{old}}, \mathbf{H}^t_{\text{new}} \in \mathbb{R}^{d \times B}$. Following the information-theoretic viewpoint in Matrix-SSL (Zhang et al., 2024), we regularize cross-covariances with a matrix cross-entropy (MCE) objective to distribute information uniformly across feature dimensions. Let $\mathbf{1} \in \mathbb{R}^B$ be the all-ones vector and define the centering matrix

$$\mathbf{H}_B = \mathbf{I}_B - \tfrac{1}{B}\mathbf{1}\mathbf{1}^\top. \tag{4}$$

For any $\mathbf{A}, \mathbf{B} \in \mathbb{R}^{d \times B}$, the zero-mean sample cross-covariance is estimated by $\frac{1}{B}\mathbf{A}\,\mathbf{H}_B\,\mathbf{B}^\top \in \mathbb{R}^{d \times d}$. We then minimize

$$\mathcal{L}_{\text{RCIP}} = \underbrace{\text{MCE}\Big(\tfrac{1}{d}\mathbf{I}_d, \ \tfrac{1}{B}\,\mathbf{H}^f\,\mathbf{H}_B\,{\mathbf{H}^f_{\text{old}}}^\top\Big)}_{\text{fusion} \leftrightarrow \text{history}} + \underbrace{\text{MCE}\Big(\tfrac{1}{d}\mathbf{I}_d, \ \tfrac{1}{B}\,\mathbf{H}^f\,\mathbf{H}_B\,{\mathbf{H}^t_{\text{new}}}^\top\Big)}_{\text{fusion} \leftrightarrow \text{current}}. \tag{5}$$

Here $\text{MCE}(P, U)$ denotes a matrix cross-entropy

$$\text{MCE}(P, U) = -\log\det\left(P^{-1/2} U P^{-1/2}\right), \tag{6}$$

which drives the cross-covariance matrix $C$ towards the target covariance $P$ and penalizes degenerate or highly anisotropic directions.

Substituting Eq.(6) into Eq.(5), where $\mathbf{U}_{f,\text{old}} = \frac{1}{B}\,\mathbf{H}_f\mathbf{H}_B\mathbf{H}^\top_{\text{old}}$ and $\mathbf{U}_{f,\text{new}} = \frac{1}{B}\,\mathbf{H}_f\mathbf{H}_B\mathbf{H}^\top_{\text{new}}$, we get

$$\begin{aligned}
\mathcal{L}_{\text{RCIP}} &= \text{MCE}\Big(\tfrac{1}{d}\mathbf{I}_d, \mathbf{U}_{f,\text{old}}\Big) + \text{MCE}\Big(\tfrac{1}{d}\mathbf{I}_d, \mathbf{U}_{f,\text{new}}\Big) \\
&= -\log\det\Big(\big(\tfrac{1}{d}\mathbf{I}_d\big)^{-1/2}\mathbf{U}_{f,\text{old}}\big(\tfrac{1}{d}\mathbf{I}_d\big)^{-1/2}\Big) - \log\det\Big(\big(\tfrac{1}{d}\mathbf{I}_d\big)^{-1/2}\mathbf{U}_{f,\text{new}}\big(\tfrac{1}{d}\mathbf{I}_d\big)^{-1/2}\Big).
\end{aligned} \tag{7}$$

Since $\big(\tfrac{1}{d}\mathbf{I}_d\big)^{-1/2} = \sqrt{d}\,\mathbf{I}_d$, the Eq.(7) can be simplified to

$$\begin{aligned}
\mathcal{L}_{\text{RCIP}} &= -\log\det\big(\sqrt{d}\,\mathbf{I}_d\,\mathbf{U}_{f,\text{old}}\,\sqrt{d}\,\mathbf{I}_d\big) - \log\det\big(\sqrt{d}\,\mathbf{I}_d\,\mathbf{U}_{f,\text{new}}\,\sqrt{d}\,\mathbf{I}_d\big) \\
&= -\log\det\big(d\,\mathbf{U}_{f,\text{old}}\big) - \log\det\big(d\,\mathbf{U}_{f,\text{new}}\big) \\
&= -\big(d\log d + \log\det\mathbf{U}_{f,\text{old}}\big) - \big(d\log d + \log\det\mathbf{U}_{f,\text{new}}\big) \\
&= -2d\log d - \log\det\mathbf{U}_{f,\text{old}} - \log\det\mathbf{U}_{f,\text{new}}.
\end{aligned} \tag{8}$$

where $2d\log d$ is a constant term. This form clearly shows that minimizing RCIP is equivalent to maximizing the log-determinant of $\mathbf{U}_{f,\text{old}}$ and $\mathbf{U}_{f,\text{new}}$, which encourages cross-temporal mutual information between the fused representation and the historical/new modalities while suppressing redundant directions. We ensure numerical stability by adding a small diagonal jitter and operating on a positive-definite matrix $\widetilde{\mathbf{U}} = \mathbf{U}\mathbf{U}^\top + \epsilon\mathbf{I}_d, \quad \epsilon = 10^{-4}$, which keeps all eigenvalues bounded away from zero, and then computing $-\log\det(\widetilde{\mathbf{U}})$ using a stable Cholesky-based routine instead of a raw determinant. Under this setup, we did not observe singularities, NaNs, or exploding values in any experiment.

**Proposition 1.** *Minimizing the matrix cross-entropy (MCE) loss is equivalent to maximizing the cross-temporal mutual information between the fused representation and historical/current modalities. The proof process is detailed in Appendix A.2.*

Based on Proposition 1, we further explore that Eq.(5) also plays a crucial role in redundancy suppression.

**Proposition 2.** *Maximizing the cross-temporal mutual information between the fused representation and the historical/current modality can eliminate (a) intra-representation redundancy inside $H_f$ and (b) cross-temporal redundancy orthogonal to shared/complementary directions.* The proof process is detailed in Appendix A.3.

Combining Proposition 1 and 2, we can conclude that minimizing Eq.(5) enables the fused representation to absorb complementary information while suppressing redundancy. Different from existing studies that adopt MCE as a static uniformity regularizer, we extend MCE to continual multi-modal clustering, thereby achieving reliable knowledge transfer and redundancy-suppressed. Detailed discussion of the differences with the prior art can be found in Appendix A.5. Intuitively, the MCE term encourages the cross-covariance matrices between $H_f$ and $(H_f^{\text{old}}, H_t^{\text{new}})$ to approach a scaled identity matrix. This means that each feature dimension contributes a similar amount of variance and that different dimensions are decorrelated. As a result, the information shared between historical and current representations is evenly distributed across many directions instead of being concentrated in a few dominant ones. This viewpoint naturally bridges feature-uniformity regularization and cross-temporal mutual information maximization in our optimization.

### 3.2 SEQUENCE-AGNOSTIC ANTI-FORGETTING STRATEGY

A typical challenge that breaks sequence-agnosticity in continual multi-modal learning is that high-quality modalities often arrive first; as more modalities accumulate, model components gradually forget the earlier high-quality information. To address this, we design a sequence-agnostic anti-forgetting mechanism with two parts: (a) cross-temporal consistency transfer to pass historical knowledge forward, and (b) quality-aware consolidation to preserve reliable signals adaptively.

**Cross-temporal Consistency Transfer (CCT).** CCT aligns current representations with their previous-moment counterparts at both the feature and cluster-distribution levels, ensuring smooth knowledge inheritance and preventing boundary drift. At step $t$, with $X_t$ modalities observed, let $(\mathbf{Z}^{(v)}, \mathbf{C}^{(v)})$ denote the current features and cluster assignments of modality $v$ ($v \in 1, \ldots, t-1$), and $(\mathbf{Z}_{\text{old}}^{(v)}, \mathbf{C}_{\text{old}}^{(v)})$ their previous counterparts. The CCT loss is defined as

$$\mathcal{L}_{\text{CCT}} = \sum_{v=1}^{t-1} \left( \left|\left| [\mathbf{Z}^{(v)}, \mathbf{C}^{(v)}] - [\mathbf{Z}^{(v)}\text{old}, \mathbf{C}_{\text{old}}^{(v)}] \right|\right|_2^2 + \text{KL}\left( \mathbf{Q} \,\|\, \mathbf{C}^{(v)} \right) \right), \tag{9}$$

The first terms enforce temporal smoothness of features and assignments, while the KL term leverages $\mathbf{Q}$ as soft supervision to stabilize clustering against drift.

**Quality-aware Historical Consolidation (QHC).** To prevent forgetting of high-quality information, QHC adaptively adjusts retention strength according to each modality's consistency with the fused semantics and applies it to cross-modal alignment. For any pair of modalities $(v, w)$, we perform contrastive learning on both feature embeddings $(\mathbf{H}^{(v)}, \mathbf{H}^{(w)})$ and cluster assignments $(\mathbf{C}^{(v)}, \mathbf{C}^{(w)})$. The contrastive loss is

$$\mathcal{L}_{\text{CL}}(a, b) = -\log \frac{\exp(\text{sim}(a, b)/\tau)}{\sum b' \in \mathcal{N}(a) \exp(\text{sim}(a, b')/\tau)}, \tag{10}$$

where $\text{sim}(\cdot, \cdot)$ denotes cosine similarity, $\tau$ is the temperature, and $\mathcal{N}(a)$ is the negative set. The overall QHC loss is

$$\mathcal{L}_{\text{QHC}} = \sum_{v=1}^{t} \left( w^{(v)} \mathcal{L}_{\text{CL}}(\mathbf{H}^{(v)}, \mathbf{H}_{\text{new}}^t) + w^{(v)}(\mathcal{L}_{\text{CL}}(\mathbf{C}^{(v)}, \mathbf{C}_{\text{new}}^t) - H(C)) \right). \tag{11}$$

where $H(C) = \sum_{i=1}^{t} \sum_{j=1}^{k} \left( \frac{1}{N} \sum_{r=1}^{N} c_{rj}^i \right) \log \left( \frac{1}{N} \sum_{r=1}^{N} c_{rj}^i \right)$ is the entropy of cluster assignments. Maximizing entropy (via the $-H$ term) discourages degenerate solutions while aligning the new modality with the historical semantic space.

**Modality-Quality Weights via MMD.** The adaptive weight $w^{(v)}$ is derived from the distributional similarity between modality features $\mathbf{H}^{(v)}$ and the fused representation $\mathbf{H}^f$. With a Gaussian kernel $k(\cdot, \cdot)$, the empirical squared MMD is

$$\text{MMD}^2(\mathbf{H}^{(v)}, \mathbf{H}^f) = \frac{1}{B^2} \sum_{i,i'=1}^{B} k\big(\mathbf{H}_i^{(v)}, \mathbf{H}_{i'}^{(v)}\big) + \frac{1}{B^2} \sum_{j,j'=1}^{B} k\big(\mathbf{H}_j^f, \mathbf{H}_{j'}^f\big) - \frac{2}{B^2} \sum_{i=1}^{B} \sum_{j=1}^{B} k\big(\mathbf{H}_i^{(v)}, \mathbf{H}_j^f\big). \tag{12}$$

We map it to a quality score and normalize:

$$s^{(v)} = \exp\big(-\text{MMD}^2(\mathbf{H}^{(v)}, \mathbf{H}^f)\big), \qquad w^{(v)} = \frac{s^{(v)}}{\sum_{u=1}^{t} s^{(u)}}. \tag{13}$$

Modalities better aligned with the fused representation obtain larger weights and thus contribute more to consolidation.

The sequence-agnostic anti-forgetting loss integrates both parts:

$$\mathcal{L}_{\text{SAAS}} = \mathcal{L}_{\text{CCT}} + \mathcal{L}_{\text{QHC}}. \tag{14}$$

Here, the $\mathcal{L}_{\text{CCT}}$ enforces temporal smoothness at feature and cluster levels, while the $\mathcal{L}_{\text{QHC}}$ term amplifies reliable modalities through quality-aware weighting. Together, they mitigate path dependence on modality sequence and enhance sequence-agnostic reliability in continual clustering.

## 3.3 OPTIMIZATION

The overall objective combines the reliable continual information propagation and sequence-agnostic anti-forgetting losses. Specifically, at each time step $t$, we integrate the newly arrived modality $\mathbf{H}^t$ with the historical fused representation $\mathbf{H}_{old}^f$, and optimize the model using

$$\mathcal{L} = \mathcal{L}_{\text{RCIP}} + \mathcal{L}_{\text{SAAS}}. \tag{15}$$

The first term ensures the reliability of the continual learning process, and the second term prevents high-quality modalities from being forgotten, thereby obtaining a stable and sequence-agnostic cluster representation. Algorithm 1 provides the optimization process of SCMC. Details of Algorithm 1 are in Appendix A.4.

## 4 EXPERIMENTS

### 4.1 EXPERIMENT SETTINGS

**Datasets.** We evaluate our method on five challenging multi-modal datasets. Details of datasets are in Appendix A.6.

**State-of-the-art Methods.** To evaluate performance, we compared our approach against four classic clustering methods (**KM** (K-Means), **Ncuts** (Normalized Cuts), **AmKM** (All-modal K-Means), and **AmNcuts** (All-modal Normalized Cuts)), three traditional multi-view clustering methods (**RMKMC** (Cai et al., 2013), **SwMC** (Nie et al., 2017), **SMVSC**(Sun et al., 2021)), six recent deep SOTA multi-modal clustering methods (**CoMVC** (Trosten et al., 2021), **MFLVC** (Xu et al., 2022), **SPDMC**(Chen et al., 2023), **DealMVC** (Yang et al., 2023), **DIVIDE** (Lu et al., 2024), and **SSLNMVC** (Yan et al., 2025)) and two existing advanced continual multi-modal clustering methods (**ProNet** (Mao et al., 2025) and **CCMVC-FSF** (Wan et al., 2024)). ProNet and CCMVC-FSF under continuous multi-modal settings. The selection of baselines followed by Wang et al. (2025a).

**Implementation Details.** Details of implementation are in Appendix A.7.

### 4.2 CLUSTERING RESULTS AND ANALYSIS

By analyzing the results shown in Table 1, we have the following observations: (1) SCMC consistently outperforms all baselines, improving ACC by 10.3%, 3.0%, 3.3%, and 15.2% on the first four multi-modal datasets, with the largest gain on 20NewsGroup. These gains stem from two key

Table 1: The comparison results of different methods on multi-modal datasets. The best results are highlighted in bold, and the second-best result is underlined.

| Method | WVU1 | | | Crosstask | | | Event | | | 20NewsGroup | | | Caltech3M | | | PBMC | | | NusWide | | |
|---|---|---|---|---|---|---|---|---|---|---|---|---|---|---|---|---|---|---|---|---|---|
| | ACC | NMI | PUR | ACC | NMI | PUR | ACC | NMI | PUR | ACC | NMI | PUR | ACC | NMI | PUR | ACC | NMI | PUR | ACC | NMI | PUR |
| KM | 0.308 | 0.372 | 0.339 | 0.451 | 0.403 | 0.484 | 0.346 | 0.208 | 0.365 | 0.221 | 0.041 | 0.223 | 0.463 | 0.313 | 0.488 | 0.381 | 0.405 | 0.641 | 0.268 | 0.167 | 0.271 |
| Ncuts (TPAMI'00) | 0.599 | 0.589 | 0.574 | 0.462 | 0.361 | 0.471 | 0.339 | 0.154 | 0.349 | 0.413 | 0.264 | 0.423 | 0.426 | 0.254 | 0.449 | 0.388 | 0.421 | 0.520 | 0.316 | 0.141 | 0.316 |
| AmKM | 0.279 | 0.251 | 0.310 | 0.464 | 0.392 | 0.489 | 0.283 | 0.114 | 0.289 | 0.209 | 0.013 | 0.213 | 0.469 | 0.315 | 0.471 | 0.311 | 0.285 | 0.610 | 0.268 | 0.152 | 0.281 |
| AmNcuts (TPAMI'00) | 0.583 | 0.610 | 0.606 | 0.338 | 0.271 | 0.367 | 0.349 | 0.199 | 0.368 | 0.612 | 0.472 | 0.612 | 0.437 | 0.255 | 0.457 | 0.401 | 0.392 | 0.594 | 0.304 | 0.161 | 0.312 |
| RMKMC (IJCAI'13) | 0.460 | 0.533 | 0.503 | 0.371 | 0.280 | 0.385 | 0.357 | 0.215 | 0.391 | 0.406 | 0.279 | 0.408 | 0.595 | 0.494 | 0.626 | 0.290 | 0.207 | 0.503 | 0.305 | 0.145 | 0.308 |
| SwMC (IJCAI'17) | 0.289 | 0.279 | 0.339 | 0.324 | 0.325 | 0.364 | 0.166 | 0.018 | 0.204 | 0.305 | 0.136 | 0.305 | 0.302 | 0.231 | 0.329 | 0.319 | 0.121 | 0.319 | 0.125 | 0.150 | 0.153 |
| SMVSC (ACM MM'21) | 0.418 | 0.101 | 0.430 | 0.491 | 0.383 | 0.495 | 0.388 | 0.249 | 0.401 | 0.598 | 0.428 | 0.606 | 0.548 | 0.433 | 0.645 | 0.468 | 0.413 | 0.626 | 0.341 | 0.172 | 0.341 |
| CoMVC (CVPR'21) | 0.423 | 0.444 | 0.442 | 0.447 | 0.419 | 0.467 | 0.427 | 0.292 | 0.452 | 0.316 | 0.104 | 0.330 | 0.541 | 0.504 | 0.586 | 0.446 | 0.428 | 0.607 | 0.430 | 0.277 | 0.430 |
| MFLVC (CVPR'22) | 0.582 | 0.613 | 0.597 | 0.596 | 0.517 | 0.596 | 0.491 | 0.369 | 0.522 | 0.658 | 0.621 | 0.658 | 0.631 | 0.566 | 0.684 | 0.336 | 0.486 | 0.637 | 0.432 | 0.267 | 0.432 |
| SPDMC (TNNLS'23) | 0.329 | 0.313 | 0.338 | 0.321 | 0.293 | 0.363 | 0.247 | 0.108 | 0.261 | 0.218 | 0.039 | 0.222 | 0.514 | 0.405 | 0.527 | 0.318 | 0.034 | 0.349 | 0.354 | 0.221 | 0.354 |
| DealMVC (ACM MM'23) | 0.552 | 0.603 | 0.571 | 0.335 | 0.294 | 0.335 | 0.509 | 0.371 | 0.532 | 0.632 | 0.571 | 0.658 | 0.595 | 0.568 | 0.542 | 0.349 | 0.418 | 0.611 | 0.370 | 0.234 | 0.374 |
| DIVIDE (AAAI'24) | 0.499 | 0.500 | 0.529 | 0.447 | 0.406 | 0.506 | 0.286 | 0.115 | 0.299 | 0.426 | 0.170 | 0.432 | 0.595 | 0.536 | 0.627 | 0.391 | 0.360 | 0.534 | 0.363 | 0.260 | 0.363 |
| SSLNMVC (TMM'25) | 0.506 | 0.569 | 0.365 | 0.586 | 0.537 | 0.588 | 0.460 | 0.381 | 0.502 | 0.602 | 0.542 | 0.612 | 0.637 | 0.591 | 0.654 | 0.297 | 0.453 | 0.205 | 0.411 | 0.268 | 0.411 |
| CCMVC-FSF (TNNLS'24) | 0.531 | 0.559 | 0.555 | 0.520 | 0.404 | 0.528 | 0.451 | 0.267 | 0.462 | 0.414 | 0.272 | 0.462 | 0.681 | 0.549 | 0.701 | 0.412 | 0.397 | 0.453 | 0.397 | 0.238 | 0.397 |
| ProNet (KBS'25) | 0.470 | 0.429 | 0.558 | 0.578 | 0.503 | 0.583 | 0.308 | 0.174 | 0.323 | 0.576 | 0.368 | 0.692 | 0.735 | 0.664 | 0.735 | 0.364 | 0.413 | 0.386 | 0.410 | 0.260 | 0.412 |
| **SCMC** | **0.686** | **0.699** | **0.697** | **0.626** | **0.548** | **0.654** | **0.542** | **0.396** | **0.575** | **0.810** | **0.668** | **0.810** | 0.729 | 0.622 | 0.729 | **0.537** | **0.489** | **0.665** | **0.448** | **0.278** | **0.451** |

Table 2: The results of ablation experiments on multi-modal datasets. The best results are highlighted in bold.

| Datasets | WVU1 | | | Crosstask | | | Event | | | 20NewsGroup | | | Caltech3M | | | PBMC | | | NusWide | | |
|---|---|---|---|---|---|---|---|---|---|---|---|---|---|---|---|---|---|---|---|---|---|
| | ACC | NMI | PUR | ACC | NMI | PUR | ACC | NMI | PUR | ACC | NMI | PUR | ACC | NMI | PUR | ACC | NMI | PUR | ACC | NMI | PUR |
| W/O $\mathcal{L}_{RCIP}$ | 0.651 | 0.677 | 0.323 | 0.614 | 0.546 | 0.647 | 0.538 | 0.399 | 0.575 | 0.770 | 0.581 | 0.770 | 0.710 | 0.600 | 0.710 | 0.328 | 0.472 | 0.612 | 0.401 | 0.213 | 0.309 |
| W/O $\mathcal{L}_{SAAS}$ | 0.151 | 0.078 | 0.152 | 0.262 | 0.180 | 0.262 | 0.281 | 0.211 | 0.282 | 0.576 | 0.431 | 0.576 | 0.354 | 0.274 | 0.354 | 0.491 | 0.441 | 0.613 | 0.274 | 0.144 | 0.274 |
| **SCMC** | **0.686** | **0.699** | **0.697** | **0.626** | **0.548** | **0.654** | **0.542** | **0.396** | **0.575** | **0.810** | **0.668** | **0.810** | **0.729** | **0.622** | **0.729** | **0.537** | **0.489** | **0.665** | **0.448** | **0.278** | **0.351** |

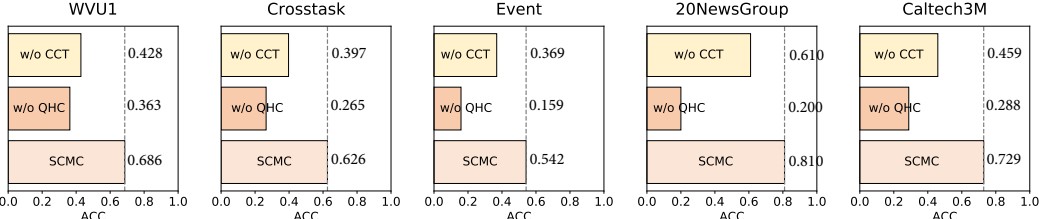

Figure 3: Ablation results of sequence-agnostic anti-forgetting (SAAS) on different datasets.

designs: (a) a reliable continual learning framework that suppresses noisy, task-irrelevant information while preserving useful signals, and (b) a sequence-agnostic anti-forgetting mechanism that selectively retains valuable historical information. (2) The performance of traditional multi-view methods is slightly better than classic clustering methods, as they can leverage complementary information across views. However, relying on shallow feature representations limits their ability to capture complex nonlinear and high-order semantics, leading to lower overall performance compared with deep clustering methods. By contrast, SCMC effectively exploits high-order semantic information across modalities to achieve superior clustering performance. (3) Deep MMC methods generally outperform classical clustering due to their ability to exploit cross-modal semantics, though SPDMC underperforms on Crosstask and 20NewsGroup for lacking discriminative representations. SCMC alleviates this via residual fusion and cross-temporal knowledge collaboration. (4) Compared with CCMVC-FSF and ProNet, SCMC shows further ACC improvements of 15.5%, 4.8%, 9.1%, and 23.4%, respectively, owing to its anti-forgetting design that preserves high-quality historical knowledge while mitigating interference from new modalities. (5) Although the results on the Caltech3M dataset are slightly lower than ProNet, we still maintain second place among all methods. In addition, our modality sequence sensitivity is more stable than ProNet in Caltech3M, as shown in Figure 5(c).

## 4.3 ABLATION STUDY

**Ablation on Model Components.** We conducted ablations on RCIP and SAAS modules. As shown in Table 2, removing any module degraded performance, with the largest drop from eliminating SAAS. This highlights its role in reliable information retention, which becomes essential as contin-

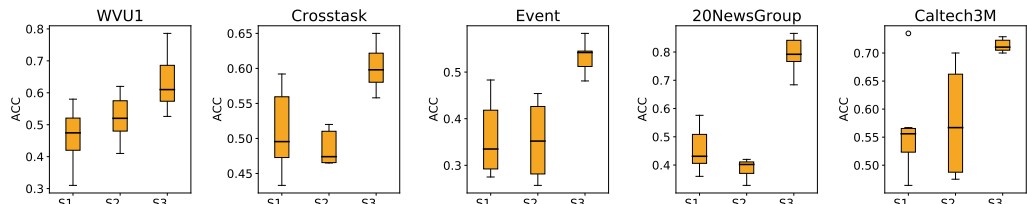

Figure 4: Boxplots of different modality sequence combinations. S1 represents the ProNet method, S2 represents the CCMVC-FSF method, and S3 represents the SCMC method. The shorter the yellow rectangle, the lower the volatility.

ual learning deepens; otherwise, the process degenerates into clustering only the latest modality. To further assess SAAS, we performed ablations within this module.

**Ablation on Sequence-agnostic Anti-forgetting.** As shown in Figure 3, the Quality-aware Historical Consolidation (QHC) module plays an important role. Specifically, on the Crosstask and 20NewsGroup datasets, removing the QHC module significantly reduces performance by 36.1% and 61%, respectively. This phenomenon shows that retaining valuable information is crucial for clustering performance.

**Ablation on MCE and QHC.** Table 5 shows that the MCE and QHC losses do not have overlapping effects. When both losses are removed simultaneously, the SCMC method performs poorly on all multi-modal datasets. However, adding either one significantly improves clustering performance, and the SCMC method exhibits optimal clustering performance when both are used simultaneously. This indicates that they cooperate and complement each other.

**Ablation on RFN and pretraining.** Table 6 demonstrates that RNF effectively guarantees the reliability of the continuous learning framework, and pre-training lays a reliable foundation for high-quality multimodal feature extraction.

### 4.4 SEQUENCE-AGNOSTIC ANALYSIS

Figure 4 shows a comparison of performance stability under different modal arrival orders. We use box plots for visualization and divide all sequence combinations into four intervals to reduce the impact of extreme maximum and minimum values. The length of the yellow rectangular area reflects the magnitude of performance fluctuations. As can be seen from the figure, on the Crosstask, Event, and Caltech3M datasets, the SCMC method has the most compact boxes, indicating that it has the lowest sequence sensitivity, which strongly verifies the effectiveness of the proposed sequence-agnostic anti-forgetting strategy. On the WVU1 and 20NewsGroup datasets, SCMC's volatility is at a moderate level, but it still significantly outperforms existing methods in terms of overall clustering accuracy, further demonstrating the advantages of the proposed reliable continuous learning framework in ensuring clustering performance. **Adversarial ranking.** We used K-means to evaluate the quality of each modality in the Caltech3M dataset, with higher accuracy indicating higher modality quality. We found that modality 3 had the highest quality, followed by modality 2, and modality 1 had the lowest quality. Figure 5(c) shows that SCMC performed very stably in the 123 and 321 rankings, while the clustering performance of baselines fluctuated significantly.

### 4.5 STATISTICAL SIGNIFICANCE ANALYSIS

**Statistical significance on baselines.** We further conduct paired $t$-tests and report effect sizes (Cohen's) to assess the statistical significance of the improvements brought by SCMC over existing multi-modal clustering methods. As illustrated in Figure 6, SCMC consistently and significantly outperforms all baselines, yielding not only smaller $p$-values but also medium-to-large effect sizes. These results indicate that the performance gains of SCMC are both statistically significant and practically meaningful, demonstrating its stronger and more reliable clustering capability compared with prior methods.

**Statistical significance on different modal sequences of existing CMC methods.** As shown in Figure 7, the y-axis denotes the negative logarithm of the $p$-value $(-\log(p))$, and the x-axis cor-

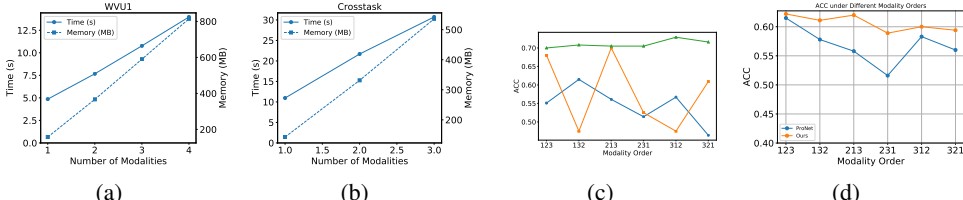

Figure 5: (a) and (b) show the time and memory cost of SCMC on the WVU1 and Crosstask datasets as the number of modalities increases. (c) represents the clustering performance of the SCMC method and existing CMC methods on the Caltech3M dataset, showing the order in which modalities arrive. (d) represents the clustering performance of SCMC and ProNet on the Caltech3M dataset under stress testing, showing the order in which modalities arrive.

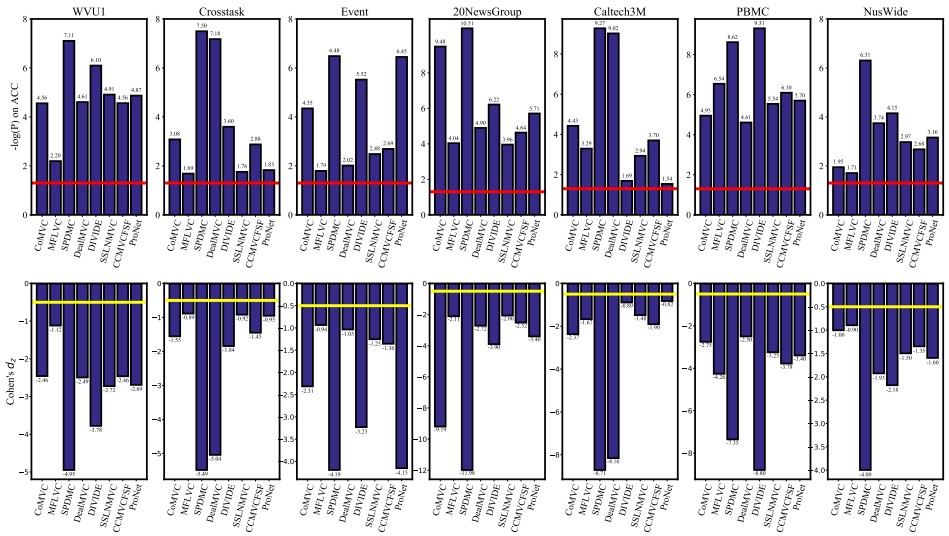

Figure 6: Statistical significance test results of multi-modal datasets. The red line indicates the significance threshold of 0.05, and the yellow line indicates the effect size threshold of 0.5.

responds to the baseline methods. The red line in each subplot marks the significance threshold $p = 0.05$, which is equivalent to $-\log(0.05) \approx 1.3$. When the negative logarithm of the $p$-value exceeds the threshold, SCMC demonstrates statistically superior performance over other methods. Across all datasets, SCMC consistently meets this criterion, with the strongest effect on WVU1, which involves four modalities. This underscores SCMC's advantage in complex multi-modal scenarios and provides clear statistical evidence of its effectiveness.

## 5 CONCLUSION

In this paper, we propose a novel sequence-agnostic continual multi-modal clustering method (SCMC). Unlike existing approaches, SCMC eliminates the strong dependency on modality arrival sequence, making it more suitable for real-world scenarios where multi-modal data streams emerge continuously. The core idea of SCMC is to achieve both reliable continual learning and robustness to modality sequence. Specifically, the reliable continual learning framework preserves information relevant to downstream tasks, while the sequence-independent anti-forgetting mechanism ensures that valuable historical knowledge is effectively maintained. Extensive experiments have verified the effectiveness and feasibility of SCMC. In future work, we aim to extend SCMC to applications involving larger numbers of modalities and more complex environments, thereby advancing the practical deployment of continual multi-modal clustering.

ETHICS STATEMENT

This paper does not involve any potential ethics issues.

REPRODUCIBILITY STATEMENT

We have made extensive efforts to ensure the reproducibility of our work. The main paper provides clear descriptions of the proposed framework, algorithmic components, and evaluation protocols. The appendix contains complete proofs of theoretical results and detailed derivations of the key propositions. For the experiments, we select public datasets and describe the experimental configuration in detail. To facilitate reproducibility, the complete source code will be released publicly upon acceptance of this paper.

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

# A APPENDIX

In the supplemental material:

## A.1 THE STATEMENT OF USING LARGE LANGUAGE MODELS

The authors confirm that no large language models were used in the preparation of this manuscript. The conceptual design, technical methodology, experimental implementation, data analysis, and manuscript writing were all conceived and carried out solely by the authors.

## A.2 PROOF OF PROPOSITION 1

**Proposition 1.** *Minimizing the matrix cross-entropy (MCE) loss is equivalent to maximizing the cross-temporal mutual information between the fused representation and historical/current modalities.*

*Proof.* Given two feature matrices $Z_1 \in \mathbb{R}^{d \times B}$ and $Z_2 \in \mathbb{R}^{d \times B}$, let the centering matrix be

$$H_B = I_B - \tfrac{1}{B}\mathbf{1}\mathbf{1}^\top. \tag{16}$$

The zero-mean cross-covariance is defined as

$$C_{12} = \tfrac{1}{B} Z_1 H_B Z_2^\top. \tag{17}$$

The matrix cross-entropy objective is

$$\mathcal{L}_{\mathrm{MCE}} = -\log\det\left(d \cdot C_{12}\right). \tag{18}$$

**Step 1 (Entropy interpretation in matrix form).** When minimizing $\mathcal{L}_{\mathrm{MCE}}$, we enforce

$$C_{12} \;\rightarrow\; \tfrac{1}{d}I_d. \tag{19}$$

This implies equal variance across all dimensions and vanishing cross-dimension correlations:

$$\mathrm{Cov}[Z_{1,i}, Z_{2,j}] = 0, \quad i \neq j, \qquad \mathrm{Var}[Z_{1,i}] = \mathrm{Var}[Z_{2,i}] = \tfrac{1}{d}. \tag{20}$$

The joint covariance matrix is

$$\Sigma_{12} = \begin{bmatrix} I_d & C_{12} \\ C_{12}^\top & I_d \end{bmatrix}. \tag{21}$$

As $C_{12} \rightarrow \tfrac{1}{d}I_d$, the joint entropy satisfies

$$H(Z_1, Z_2) = \tfrac{1}{2}\log\det(2\pi e \cdot \Sigma_{12}) \;\rightarrow\; \min. \tag{22}$$

**Step 2 (Mutual information definition).** By definition,

$$I(Z_1; Z_2) = H(Z_1) + H(Z_2) - H(Z_1, Z_2). \tag{23}$$

Since $H(Z_1)$ and $H(Z_2)$ are fixed by the marginal distributions, minimizing $H(Z_1, Z_2)$ directly implies

$$I(Z_1; Z_2) \rightarrow \max. \tag{24}$$

**Step 3 (Link between MI and cross-covariance).** For zero-mean Gaussian variables,

$$I(Z_1; Z_2) = -\tfrac{1}{2} \log \det(I - C_{12} C_{12}^\top). \tag{25}$$

Let the singular value decomposition be

$$C_{12} = U \operatorname{diag}(\rho_1, \ldots, \rho_d) V^\top, \tag{26}$$

where $\rho_i$ are the canonical correlations with $\rho_i \in [0, 1)$. Then

$$I(Z_1; Z_2) = \tfrac{1}{2} \sum_{i=1}^{d} -\log(1 - \rho_i^2). \tag{27}$$

Since the function $f(\rho) = -\log(1 - \rho^2)$ is strictly increasing in $\rho \in [0, 1)$, we conclude

$$\rho_i \uparrow \; \Rightarrow \; I(Z_1; Z_2) \uparrow. \tag{28}$$

**Step 4 (Monotonic relation between MCE and MI).** On the other hand,

$$\mathcal{L}_{\mathrm{MCE}} = -d \log d - \log \det(C_{12}). \tag{29}$$

Because

$$\log \det(C_{12}) = \sum_{i=1}^{d} \log \rho_i, \tag{30}$$

minimizing $\mathcal{L}_{\mathrm{MCE}}$ is equivalent to maximizing $\sum_{i=1}^{d} \log \rho_i$.

Combining Step 3 and Step 4, both $\sum_{i=1}^{d} \log \rho_i$ and $I(Z_1; Z_2) = \tfrac{1}{2} \sum_i -\log(1 - \rho_i^2)$ are strictly increasing in each $\rho_i$, and thus

$$\min \mathcal{L}_{\mathrm{MCE}} \; \Leftrightarrow \; \max I(Z_1; Z_2). \tag{31}$$

Therefore, minimizing the matrix cross-entropy loss drives the cross-covariance towards isotropy, which in turn maximizes the cross-temporal mutual information.

$$\square$$

### A.3 PROOF OF PROPOSITION 2

**Proposition 2.** *Maximizing the cross-temporal mutual information between the fused representation and the historical/current modality can eliminate (a) intra-representation redundancy inside $H_f$ and (b) cross-temporal redundancy orthogonal to shared/complementary directions.*

*Proof.* Let $H_f \in \mathbb{R}^{d \times B}$ be the zero-mean fused representation at step $t$ with covariance $\Sigma_f = \frac{1}{B} H_f H_f^\top$ and correlation $R_f = D^{-1/2} \Sigma_f D^{-1/2}$ where $D = \operatorname{diag}(\Sigma_f)$. For any cross-temporal pair $(Z_1, Z_2) \in \{(H_f, H_{\mathrm{old}}^f), (H_f, H_{\mathrm{new}}^t)\}$, the CTKC objective minimizes the matrix cross-entropy $\mathcal{L}_{\mathrm{MCE}} = -\log \det(d \cdot C_{12})$ with $C_{12} = \frac{1}{B} Z_1 H_B Z_2^\top$ and $H_B = I_B - \frac{1}{B} \mathbf{1} \mathbf{1}^\top$. Under per-view whitening, minimizing $\mathcal{L}_{\mathrm{MCE}}$ is equivalent to (i) enforcing isotropy $R_f \rightarrow I_d$ and (ii) maximizing a strictly monotone surrogate of the cross-temporal mutual information via $\sum_{i=1}^{d} \log \rho_i$ where $\{\rho_i\}$ are the canonical correlations.

*Part (a): Intra-representation redundancy.* Define Total Correlation (TC) of $H_f$ by

$$\mathrm{TC}(H_f) = D_{\mathrm{KL}}\big(p(H_f) \,\big\|\, \textstyle\prod_{i=1}^{d} p(H_{f,i})\big) = \textstyle\sum_{i=1}^{d} H(H_{f,i}) - H(H_f). \tag{32}$$

For zero-mean Gaussian $H_f$ with fixed marginal variances, $H(H_{f,i})$ are constants and the identity

$$\mathrm{TC}(H_f) = -\tfrac{1}{2} \log \det(R_f) \tag{33}$$

holds. CTKC's uniformity/isotropy constraint yields $R_f \to I_d$, hence

$$R_f \to I_d \quad \implies \quad \mathrm{TC}(H_f) \to 0, \tag{34}$$

i.e., dimensions of $H_f$ become decorrelated and equally utilized, and internal (intra-$H_f$) redundancy vanishes.

*Part (b): Cross-temporal redundancy.* Let the SVD of the zero-mean cross-covariance be

$$C_{12} = U \operatorname{diag}(\rho_1, \ldots, \rho_d) V^\top, \qquad \rho_i \in [0, 1). \tag{35}$$

From the MCE form,

$$\mathcal{L}_{\mathrm{MCE}} = -d \log d - \log \det(C_{12}) \quad \Longleftrightarrow \quad \max \log \det(C_{12}) = \max \sum_{i=1}^{d} \log \rho_i. \tag{36}$$

Decompose $Z_1$ into a component aligned with $Z_2$ and an orthogonal (redundant) component:

$$Z_1 = Z_1^{\parallel} + Z_1^{\perp}, \qquad Z_1^{\perp} \in \ker(C_{12}^\top), \qquad C_{12} Z_1^{\perp} = 0. \tag{37}$$

Then $\{\rho_i\}$ are singular values determined solely by the cross-correlation on the shared subspace spanned by $Z_1^{\parallel}$. Because $\sum_i \log \rho_i$ is strictly increasing in each $\rho_i$ and insensitive to $Z_1^{\perp}$, gradient ascent on $\sum_i \log \rho_i$ necessarily

$$\text{amplifies } Z_1^{\parallel} \quad \text{and} \quad \text{shrinks } Z_1^{\perp}, \tag{38}$$

i.e., energy is shifted from orthogonal (non-informative w.r.t. $Z_2$) directions into shared/complementary directions that increase cross-temporal correlation. Equivalently, components that do not contribute to cross-temporal MI are suppressed.

In summary, combining (a) and (b),

$$\underbrace{R_f \to I_d}_{\text{removes intra-}H_f \text{ redundancy}} \quad \text{and} \quad \underbrace{\sum_i \log \rho_i \uparrow}_{\text{suppresses cross-temporal redundancy}}, \tag{39}$$

so CTKC eliminates redundancy both within $H_f$ and across time $H_{\mathrm{old}}^f$ and $H_{\mathrm{new}}^t$.

$\square$

## A.4 Summary of the proposed SCMC algorithm

Algorithm 1 illustrates the pseudocode of SCMC, detailing the process of reliable continual learning and sequence-insensitive anti-forgetting.

---
**Algorithm 1** The proposed algorithm
---
**Input:** Multi-modal inputs $\{X^t\}_{t=1}^m$; number of clusters $y$; learning rate $\gamma$.
**Output:** Final cluster assignments.
 1: Initialize the neural network parameters $\{\theta^i\}_{i=1}^m$.
 2: **while** The arrival of a new modality **do**
 3:    **repeat**
 4:       Obtaining latent features through the autoencoder network.
 5:    **until** converge
 6:    **repeat**
 7:       Extract high-level modality representations $\{H^t\}_{t=1}^m$ by feature extractor.
 8:       Compute reliable continual information propagation loss by Eq. (5).
 9:       Compute the sequence-agnostic anti-forgetting strategy by Eq. (14).
 10:      Optimize the overall loss Eq. (15) by the Adam optimizer and back-propagate loss.
 11:    **until** converge
 12: **end while**
 13: **return** Obtaining the final clustering result.
---

## A.5 DIFFERENCES IN MATRIX CROSS-ENTROPY FROM PRIOR ART

Existing work typically applies matrix cross-entropy in self-supervised learning as a uniformity regularizer to prevent feature collapse within static unimodal. In contrast, our approach extends MCE to the continual multi-modal clustering setting, where it explicitly models cross-temporal interactions between fused, historical, and newly arrived representations. As shown in Proposition 1 and 2, minimizing our MCE objective is equivalent to maximizing cross-temporal mutual information, thereby ensuring reliable knowledge transfer and redundancy suppression. Unlike prior use of MCE as a static uniformity constraint, our formulation turns it into a cross-temporal reliability module, enabling robust knowledge retention and sequence-agnostic clustering.

## A.6 DATASET INFORMATION

We evaluate our method on five challenging multi-modal datasets. **WVU1** (Ramagiri et al., 2011) dataset is derived from the WVU action data, containing videos from non-adjacent views (1, 3, 5, 7). Features are extracted using the Harris3-D detector and STIP with HoG/HoF descriptors. **Event** (Li & Fei-Fei, 2007) dataset has 1,579 images from eight sports events with three features (Color Attention, SIFT, TPLBP), posing challenges due to background variation. **Crosstask** (Zhukov et al., 2019) dataset includes 4,700 instructional videos of 83 tasks. We select 2,600 samples grouped into 9 categories, each with video, audio, and text modalities. **20NewsGroup** [1] contains 500 documents from 20NG, each preprocessed into three views. **Caltech3M** (Fei-Fei et al., 2007) dataset has 1,400 images across 7 classes with WM, CENTRIST, and LBP features. **PBMC** (Lin et al., 2022) consists of 2,585 human peripheral blood mononuclear cell samples, each described by two modalities: mRNA and ATAC profiles. **NusWide** (Chua et al., 2009) is a multi-modal image dataset with 20,000 images from 8 categories, where each instance is paired with two types of information: image and textual annotations.

## A.7 IMPLEMENTATION DETAILS

The experiments were carried out using the PyTorch 3.8.0 framework, running on a Windows 10 workstation equipped with an NVIDIA GeForce RTX 3090 GPU and 64GB of RAM. For our approach, the encoder-decoder model was trained for a total of 100 epochs, which facilitated the subsequent feature extraction process. The Adam optimizer was used for optimization during training. In addition to our method, we evaluated other SOTA methods. For these methods, we leveraged their open-source code and followed the recommended configurations specified by the original authors to ensure a fair comparison. Results were assessed by ACC, NMI, and PUR, where higher values indicate better performance.

For all CMC methods, we tuned hyperparameters in the same continual setting, using shared search ranges for key parameters (learning rate $1 \times 10^{-4}, 3 \times 10^{-4}, 1 \times 10^{-3}, 3 \times 10^{-3}$, batch size 64, 128, 256) and exploring method-specific coefficients around values recommended in the original papers.

## A.8 STRESS TEST

To further probe the robustness of our sequence-agnostic claim, we design a stress-test on Caltech3M that explicitly targets modality quality and order. We first estimate the intrinsic quality of each modality by running K-means on each single view and ranking them by clustering ACC. We then take the best modality (third modality) in this ranking and deliberately degrade it by randomly setting 50% of its feature dimensions to zero for all samples, thereby injecting substantial noise and redundancy into the view that is normally most informative.

As shown in Figure. 5 (d), ProNet exhibits significant sequence sensitivity, with accuracy dropping noticeably when the degraded mode appears in the middle (order 231). In contrast, our method's clustering performance does not fluctuate significantly, indicating that our proposed method remains stable even when the most informative mode is severely damaged and its position in the sequence changes.

---

[1] http://lig-membres.imag.fr/grimal/data.html.

## A.9 SCALABILITY ANALYSES

Figure. 5 (a) and (b) illustrate the scalability of SCMC on the WVU1 and Crosstask datasets in terms of wall-clock time and peak memory usage as the number of modalities increases. We observe that both time and memory grow approximately linearly with the arrival of new modalities, and the increments between successive modality counts remain modest. This indicates that SCMC can accommodate additional modalities without incurring prohibitive computational or memory overhead, thereby demonstrating good scalability in practical continual multi-modal scenarios.

## A.10 TIME AND SPACE COMPLEXITY ANALYSIS

From Table 3, we observe that on both Crosstask and Caltech3M, SCMC uses a parameter scale comparable to other deep multi-modal methods and is even smaller than the heaviest baseline (e.g., SPDMC), showing that its gains do not rely on over-parameterization. Its wall-clock time per training step is in the same range as strong competitors and noticeably lower than several deep methods with more complex fusion schemes. Although SCMC requires slightly more peak memory, the overhead is modest and easily handled by a single commodity GPU. Overall, SCMC delivers superior clustering accuracy and sequence robustness while retaining competitive time and space complexity, offering a favorable efficiency–performance trade-off.

Table 4 further analyzes the cost of individual components. Across all variants, the parameter count is almost unchanged ($\approx$ 22.3M), indicating that RFN, MCE, CCT, and QHC mainly affect computation rather than model size. Removing CCT and/or QHC reduces per-step time and memory only marginally (a few milliseconds and several MB), whereas it significantly degrades accuracy and increases order sensitivity. Hence, the full, sequence-robust SCMC is only slightly heavier than its ablations but crucial for overall effectiveness and robustness.

## A.11 EXPERIMENTAL DATA IN THE PAPER

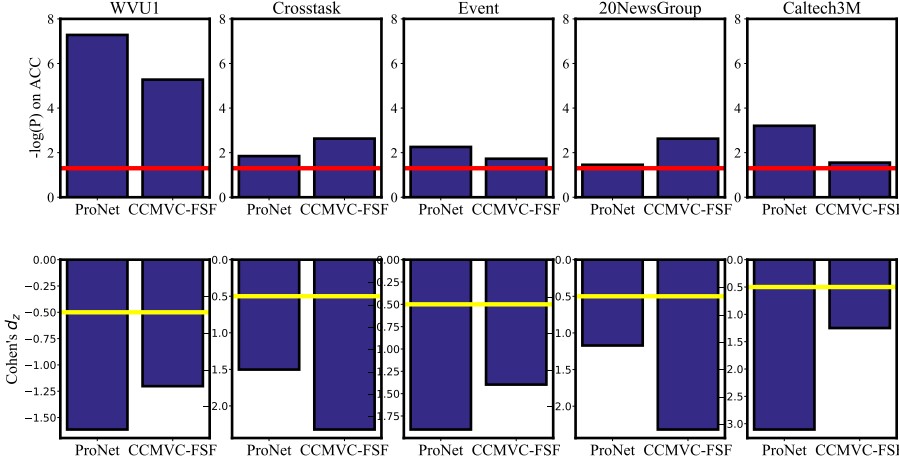

Figure 7: Statistical significance test results of different modality sequence combinations. The y-axis shows the negative logarithm of the p-value (-log($p$)), and the red line indicates the significance threshold of 0.05, the yellow line indicates the effect size threshold of 0.5.

## A.12 DIFFERENCES BETWEEN SCMC AND EXISTING CMC METHODS

Compared with existing continual multi-modal clustering methods such as ProNet and CCMVC-FSF, SCMC differs in three key aspects. 1. At the representation level, RFN and CTKC explicitly build a high-rank fused basis and regulate cross-temporal covariance through the MCE objective, whereas ProNet and CCMVC-FSF mainly rely on graph propagation or filtered structural fusion without explicitly modeling the reliability of historical–current interactions. 2. At the anti-forgetting

| Method | Crosstask | | | Caltech3M | | |
|---|---|---|---|---|---|---|
| | Params (M) | Wall-clock (ms) | Memory (MB) | Params (M) | Wall-clock (ms) | Memory (MB) |
| CoMVC | 1.583 | 9.763 | 40.77 | 1.836 | 10.402 | 44.16 |
| MFLVC | 14.463 | 46.138 | 452.99 | 14.956 | 57.401 | 359.59 |
| SPDMC | 31.921 | 10.135 | 441.81 | 33.800 | 8.202 | 435.46 |
| DealMVC | 14.871 | 32.718 | 479.44 | 10.877 | 19.351 | 248.29 |
| DIVIDE | 25.781 | 41.771 | 451.70 | 20.170 | 20.771 | 301.71 |
| SSLNMVC | 23.354 | 80.990 | 493.69 | 24.710 | 45.656 | 505.99 |
| ProNet | 1.086 | 28.055 | 83.63 | 1.529 | 52.201 | 86.92 |
| CCMVC-FSF | 10.212 | 5.434 | 102.21 | 9.221 | 4.544 | 98.71 |
| Ours | 22.310 | 22.997 | 537.52 | 22.779 | 23.715 | 554.02 |

Table 3: Parameter count, wall-clock time per training step, and peak GPU memory usage on Crosstask and Caltech3M.

| Datasets | Crosstask | | | Caltech3M | | |
|---|---|---|---|---|---|---|
| | Params (M) | Wall-clock (ms) | Memory (MB) | Params (M) | Wall-clock (ms) | Memory (MB) |
| W/O RFN | 22.31 | 21.624 | 536.94 | 22.779 | 22.736 | 553.44 |
| W/O MCE | 22.31 | 21.619 | 537.52 | 22.779 | 22.502 | 554.02 |
| W/O RFN+MCE | 22.31 | 21.144 | 536.94 | 22.779 | 21.590 | 553.44 |
| W/O CCT | 22.31 | 22.725 | 537.52 | 22.779 | 22.855 | 554.02 |
| W/O QHC | 22.31 | 16.501 | 537.52 | 22.779 | 17.501 | 554.02 |
| W/O CCT+QHC | 22.31 | 15.698 | 534.63 | 22.779 | 16.259 | 551.16 |

Table 4: Parameter count, wall-clock time per training step, and peak GPU memory usage for ablation variants on Crosstask and Caltech3M.

level, our SAAS module jointly constrains feature/cluster consistency and quality-aware contrastive alignment, while prior methods typically treat all modalities equally or only replay buffered structures, making them more vulnerable to low-quality views. 3. SCMC explicitly targets sequence-agnosticity by combining these architectural and loss-level designs, which is empirically validated in our sequence-sensitivity and significance analyses.

Table 5: Ablation study of MCE and QHC on multi-modal datasets.

| Datasets | WVU1 | | | Crosstask | | | Event | | | 20NewsGroup | | | Caltech3M | | | PBMC | | | Nuswide | | |
|---|---|---|---|---|---|---|---|---|---|---|---|---|---|---|---|---|---|---|---|---|---|
| | ACC | NMI | PUR | ACC | NMI | PUR | ACC | NMI | PUR | ACC | NMI | PUR | ACC | NMI | PUR | ACC | NMI | PUR | ACC | NMI | PUR |
| W/O MCE+QHC | 0.103 | 0.007 | 0.103 | 0.167 | 0.059 | 0.167 | 0.158 | 0.023 | 0.158 | 0.200 | 0.011 | 0.200 | 0.144 | 0.003 | 0.144 | 0.395 | 0.007 | 0.397 | 0.154 | 0.008 | 0.154 |
| W/O MCE | 0.611 | 0.604 | 0.620 | 0.567 | 0.453 | 0.567 | 0.525 | 0.384 | 0.561 | 0.782 | 0.574 | 0.782 | 0.696 | 0.598 | 0.708 | 0.416 | 0.403 | 0.637 | 0.414 | 0.237 | 0.302 |
| W/O QHC | 0.140 | 0.070 | 0.140 | 0.212 | 0.109 | 0.214 | 0.194 | 0.050 | 0.208 | 0.334 | 0.128 | 0.334 | 0.311 | 0.248 | 0.317 | 0.327 | 0.302 | 0.486 | 0.234 | 0.104 | 0.239 |
| ALL | **0.686** | **0.699** | **0.697** | **0.626** | **0.548** | **0.654** | **0.542** | **0.396** | **0.575** | **0.810** | **0.668** | **0.810** | **0.729** | **0.622** | **0.729** | **0.537** | **0.489** | **0.665** | **0.448** | **0.278** | **0.351** |

Table 6: Ablation of RFN and pretraining on multi-modal datasets.

| Variant | WVU1 | | | Crosstask | | | Event | | | 20NewsGroup | | | Caltech3M | | | PBMC | | | Nuswide | | |
|---|---|---|---|---|---|---|---|---|---|---|---|---|---|---|---|---|---|---|---|---|---|
| | ACC | NMI | PUR | ACC | NMI | PUR | ACC | NMI | PUR | ACC | NMI | PUR | ACC | NMI | PUR | ACC | NMI | PUR | ACC | NMI | PUR |
| W/O RFN | 0.595 | 0.603 | 0.611 | 0.569 | 0.504 | 0.585 | 0.454 | 0.326 | 0.462 | 0.510 | 0.386 | 0.532 | 0.630 | 0.556 | 0.630 | 0.333 | 0.303 | 0.559 | 0.406 | 0.221 | 0.309 |
| W/O Pre | 0.555 | 0.549 | 0.559 | 0.586 | 0.514 | 0.598 | 0.464 | 0.339 | 0.510 | 0.560 | 0.354 | 0.560 | 0.625 | 0.581 | 0.629 | 0.412 | 0.374 | 0.535 | 0.372 | 0.215 | 0.383 |
| ALL | **0.686** | **0.699** | **0.697** | **0.626** | **0.548** | **0.654** | **0.542** | **0.396** | **0.575** | **0.810** | **0.668** | **0.810** | **0.729** | **0.622** | **0.729** | **0.537** | **0.489** | **0.665** | **0.448** | **0.278** | **0.351** |

Table 7: Notations.

| Symbol | Description |
|---|---|
| $X = \{X_1, \ldots, X_T\}$ | Set of $T$ input modalities (views) in the continual stream |
| $X_t$ | Data of the modality that arrives at step $t$ |
| $Z_t$ | Latent representation of $X_t$ learned by the autoencoder |
| $H_t = f_\theta(Z_t)$ | High-level feature of modality $t$ after the mapping network |
| $C_t = g(H_t)$ | Cluster assignment (soft distribution) of modality $t$ |
| $H_f^{\text{old}}$ | Historical fused representation before observing the current modality |
| $H_t^{\text{new}}$ | High-level feature of the newly arrived modality at step $t$ |
| $H_f$ | Updated fused representation after cross-temporal knowledge collaboration |
| $M \in \mathbb{R}^{d \times B}$ | Concatenated multi-modal representation input to RFN |
| $d$ | Feature dimension of $H_t$ / $H_f$ |
| $B$ | Mini-batch size (number of samples in a batch) |
| $K$ | Number of clusters |
| $\text{MCE}(\cdot, \cdot)$ | Matrix cross-entropy regularizer (log-det of whitened covariance) |
| $\text{MMD}^2(\cdot, \cdot)$ | Squared maximum mean discrepancy between two representations |
| $w^{(v)}$ | Quality-aware weight of modality $v$ (softmax over $-\text{MMD}^2$ values) |
| $\mathcal{L}_{\text{RCIP}}$ | Reliable Continual Information Propagation (MCE-based) loss |
| $\mathcal{L}_{\text{CCT}}$ | Cross-Temporal Consistency Transfer loss |
| $\mathcal{L}_{\text{QHC}}$ | Quality-aware Historical Consolidation loss |
| $\mathcal{L}_{\text{SAAS}}$ | Sequence-agnostic anti-forgetting loss, $\mathcal{L}_{\text{SAAS}} = \mathcal{L}_{\text{CCT}} + \mathcal{L}_{\text{QHC}}$ |

