# OpenReview forum: "Sequence-agnostic Continual Multi-modal Clustering"
_ICLR.cc/2026/Conference — Submitted to ICLR 2026_

### Official Review · Reviewer_vdaj · 2025-10-28

**Soundness:** 3
**Presentation:** 3
**Contribution:** 4
**Rating:** 8
**Confidence:** 5

**Summary:**

This paper proposes a sequence-agnostic method for continuous multi-modal clustering, Sequence-Agnostic Continual Multi-Modal Clustering (SCMC). It aims to address two core issues in existing continuous multi-modal clustering: the unreliable fusion of historical and new modal information, and the strong dependence of clustering performance on the order of modal input. The paper first analyzes the shortcomings of existing methods in fusing noise and forgetting high-quality modalities, and constructs a reliable continuous information propagation framework. Through a residual fusion network and a cross-temporal knowledge collaboration mechanism, it achieves bidirectional information filtering between new and old modalities, thereby enhancing complementarity and suppressing redundancy. Subsequently, the authors designed a sequence-agnostic anti-forgetting strategy, including cross-temporal consistency transfer and quality-aware history integration, enabling the model to retain the discriminative knowledge of early high-value modalities when new modalities arrive.

**Strengths:**

1.This method is highly innovative and original. The paper is rigorously structured, logically coherent, and systematically comprehensive, with well-reasoned theoretical and experimental demonstrations. It represents a significant contribution to the research of multi-modal clustering.

2.SCMC explicitly solved the problem of "modal sequence sensitivity" in continuous multi-modal clustering for the first time and proposed a "sequence-independent" learning mechanism, which has important theoretical and significant value.

3.The proposed residual fusion network structurally retains the previously learned discriminative subspace and achieves smooth update of new and old modalities through up-down sampling residual blocks, suppressing update bias and improving feature stability.

4.The cross-temporal knowledge collaboration module introduces matrix cross-entropy into the continuous learning scenario, forming interpretable mutual information maximization and redundancy minimization goals; its theoretical proof (Propositions 1 and 2) is relatively complete, providing a mathematical basis for the reliability of information flow.

**Weaknesses:**

1.The description of the residual fusion network in the paper is not clear enough, and its specific role in information preservation and fusion has not been fully revealed.

2.The relationship between MCE loss and mutual information is not fully explained in the manuscript.

3.The authors are advised to further analyze and elaborate on the differences between the proposed method and existing continuous multi-modal methods.

4.Minor issues：The authors are advised to systematically proofread the some formatting issues in the paper. For example, the abbreviation "residual fusion network (RFN)" is inconsistently formatted in some sections. Furthermore, as one of the core loss terms, "Matrix Cross-Entropy (MCE)" is recommended to provide a complete and clear formula definition upon its first appearance.

**Questions:**

1.The paper's description of the Residual Fusion Network (RFN) is not clear enough, and its role in information preservation and fusion is not fully revealed. The authors are advised to further clarify how the RFN preserves the learned discriminative subspace through the residual channel and forms a high-rank global basis in the fusion process.

2.The authors are advised to further clarify in the Methods section how modal sequence-agnostic is achieved through the synergy between the model structure and the optimization objective.

3.The "Matrix Cross-Entropy (MCE)" loss is introduced in Equation (5), but the main text does not provide an intuitive understanding of its equivalence with mutual information (MI), and only provides a mathematical derivation in the Appendix. This arrangement makes it difficult for readers to directly understand the core motivation of the MCE loss and its information-theoretic significance when reading the main text. The authors are advised to add an intuitive explanation in the main text to explain how MCE plays a role in bridging feature uniformity and mutual information maximization in the optimization process.

---

> ### Author Response · Authors · 2025-11-20
> **Response to Reviewer vdaj**
>
> Thank you for this thoughtful comment.
> ***
> **Weakness 1 and Question 1**
> \
> \
> The residual path acts as an orthogonal projection that retains the previously learned discriminative subspace, ensuring that new modality updates only affect the complementary orthogonal subspace. During each iteration, the residual fusion expands the global basis by integrating new modality-specific components that are linearly independent from the historical subspace, thus forming a high-rank representation space capable of accommodating incremental semantic diversity.
>
> **Revision:**
>
> We have revised **Section 3.1 (Reliable Continual Information Propagation Framework)** to clearly introduce residual fusion networks.
>
> ***
> **Weakness 2 and Question 3**
> \
> \
> The MCE term encourages the cross-covariance matrices between $H_f$ and $(H_f^{\text{old}}, H_t^{\text{new}})$ to approach a scaled identity matrix. This means that each feature dimension contributes a similar amount of variance and that different dimensions are decorrelated. As a result, the information shared between historical and current representations is evenly distributed across many directions instead of being concentrated in a few dominant ones. This viewpoint naturally bridges feature-uniformity regularization and cross-temporal mutual information maximization in our optimization.
>
> **Revision:**
>
> In the revised manuscript, we have added **Section 3.1 (Cross-Temporal Knowledge Collaboration)** to clearly explain the relationship between MCE loss and mutual information.
>
> ***
> **Weakness 3**
> \
> \
> Compared with existing continual multi-modal clustering methods such as ProNet (Mao et al., 2025) and CCMVC-FSF (Wan et al., 2024), SCMC differs in three key aspects.
>
> 1. At the representation level, RFN and CTKC explicitly build a high-rank fused basis and regulate cross-temporal covariance through the MCE objective, whereas ProNet and CCMVC-FSF mainly rely on graph propagation or filtered structural fusion without explicitly modeling the reliability of historical–current interactions.
>
> 2. At the anti-forgetting level, our SAAS module jointly constrains feature/cluster consistency and quality-aware contrastive alignment, while prior methods typically treat all modalities equally or only replay buffered structures, making them more vulnerable to low-quality views.
>
> 3. SCMC explicitly targets sequence-agnosticity by combining these architectural and loss-level designs, which is empirically validated in our sequence-sensitivity and significance analyses.
>
>
> **Revision:**
>
> We have added a discussion on differences between SCMC and existing CMC methods in **Appendix A.10**.
> ***
> **Weakness 4**
> \
> \
> In the revision, we have:
>
> 1. Systematically proofread the manuscript to correct typos and formatting issues.
>
> 2. Unified the abbreviation ``residual fusion network (RFN)'' throughout the paper.
>
> 3. Provided a complete formula definition of MCE upon its first appearance in Eq.~(5).
>
> 4. Corrected minor notational issues such as writing the activation function as $\sigma(\cdot)$ instead of ``$\sigma = (\cdot, \cdot)$''.
>
>
> ***
> **Question 2**
> \
> \
> RFN maintains a unified fused representation $H_f$ that incrementally aggregates all seen modalities irrespective of arrival order; the CCT loss enforces each new modality to be consistent with the historical features and cluster assignments derived from $H_f$, while QHC, via quality-aware weights $w^{(v)}$, prioritizes modalities that are well aligned with this fused space. As a result, optimization converges to a common fused semantic space dominated by high-quality modalities rather than recency, yielding sequence-agnostic clustering behavior.
>
> ***
> The above revisions have been marked in blue in the revised version.
>
> If your concerns have been addressed, could you please help raise the score. If you have any other concerns, please let us know, and we will try our best to address them. Thanks.

---

> > ### Comment · Reviewer_j6TL · 2025-11-26
> >
> > Thanks for addressing my questions. I have adjusted my rating accordingly.

---

> > > ### Author Response · Authors · 2025-11-26
> > >
> > > Dear Reviewer  j6TL,
> > >
> > > Thank you for your constructive feedback and for updating your rating. We truly appreciate your time and insights.
> > >
> > > Best regards,
> > > Authors of paper 4298

---

### Official Review · Reviewer_j6TL · 2025-10-31

**Soundness:** 2
**Presentation:** 3
**Contribution:** 3
**Rating:** 4
**Confidence:** 4

**Summary:**

This submissions introduces a framework for Continual Multi-Modal Clustering that explicitly addresses the challenge of sequence sensitivity in modality arrival. The method combines a Residual Fusion Network for reliable continual information propagation, a Cross-Temporal Knowledge Collaboration mechanism to filter redundant information between historical and new modalities, and a Sequence-Agnostic Anti-Forgetting Strategy that uses cross-temporal consistency transfer and quality-aware consolidation.

**Strengths:**

1/ This submission addresses the sequence sensitivity problem in continual multi-modal clustering, an under explored but practically important challenge in streaming, multi-view data environments.

2/ The residual fusion and cross-temporal information filtering are well motivated and supported by information-theoretic derivations that enhance the paper’s credibility.

3/ The proposed SCMC integrates residual fusion with information-theoretic regularization and cross-temporal consistency, showing careful theoretical grounding.

4/The method is evaluated against a wide range of baselines, including both traditional clustering and recent continual multi-modal methods, with significant improvements.

**Weaknesses:**

1/ While the framework is technically sound, the presentation suffers from heavy notation and dense exposition. A clearer pseudocode and a concise diagram mapping each loss term to modules in Figure 2 would make the methodology easier to follow.

2/ The experiments rely on relatively small datasets. It would strengthen the claims to test SCMC on larger-scale or real-world streaming datasets, e.g., MM-IMDB, CMU-MOSI, or visual-linguistic datasets. Scalability analyses (time and memory cost vs. number of modalities) are also missing.

3/ The current ablations focus mainly on removing modules. However, the interaction between RCIP and SAAS could be explored, e.g., does the anti-forgetting still work without residual fusion? Additionally, since a pretraining step is used, it is unclear how much improvement arises from pretraining itself versus continual adaptation.

4/ The use of both MCE-based cross-temporal mutual information and contrastive QHC losses may lead to overlapping optimization effects. Clarifying their distinct roles or showing complementarity empirically would enhance methodological clarity.

**Questions:**

Refer to the weakness session.

---

> ### Author Response · Authors · 2025-11-20
> **Response to Reviewer j6TL**
>
> We greatly appreciate the reviewer’s thoughtful feedback and recognition of the contributions of our work.
> ***
> **Weakness 1**
> \
> \
> In the revised version, we modified the original pseudocode algorithm table to make it clearer and easier to understand. In Figure 2, we explicitly map each loss term to its corresponding module to make it clearer. We added a notation that covers the main symbolic representations in the paper and further simplified the symbolic representations by reusing fewer symbols.
>
> **Revisions:**
>
> 1. In **Section 3 (Problem Formulation)**, we added a clear notation.
>
> 2. In **Appendix A.4 (Summary of the proposed SCMC algorithm)**, we added a clear pseudocode.
>
> 3. In **Section 3 (Overall Framework)**, we clearly labeled the correspondence between the loss and the module in Figure 2.
> ***
> **Weakness 2**
> \
> \
> In the revised version, we added a large-scale visual-linguistic dataset (Nuswide) with 20,000 samples and a real-world streaming medical dataset (PBMC). We then conducted detailed experimental validation and analysis on these datasets. The experimental results also validated the reliability and effectiveness of SCMC. Furthermore, we performed scalability analyses on the WVU1 and crosstask datasets.
>
> **Revisions:**
>
> 1. In **Section 4 (Datasets)**, we added a large-scale visual-linguistic dataset and a real-world streaming dataset.
>
> 2. In **Appendix A.9 (Scalability analyses)**, we added scalability analyses.
> ***
> **Weakness 3**
> \
> \
> In the revised version, we added ablation experiments to remove residual fusion, further validating the effectiveness of the reliable continuous learning framework. Additionally, we added ablation experiments to remove pre-training, validating the effectiveness of the pre-training design and providing a more intuitive observation of the performance differences brought about by pre-training.
>
> **Revision:**
>
> In **Section 4.3**, we added an ablation on RFN and pretraining.
> ***
> **Weakness 4**
> \
> \
> In the revision, we clarify that RCIP (MCE) and QHC operate at different levels and are complementary rather than redundant.
>
> 1. The MCE-based RCIP loss is a matrix-level distributional objective on cross-covariances between the fused representation and historical/new modalities, shaping the global geometry of the shared space by enhancing cross-temporal dependence and suppressing redundant directions.
>
> 2. In contrast, the QHC contrastive losses act at the instance/cluster level, are gated by MMD-based quality weights, and focus on quality-aware anti-forgetting by preserving high-quality modalities and stabilizing cluster boundaries across modality orders.
>
> To substantiate this, we add an ablation with (1) both removed, (2) w/o MCE, (3) w/o QHC, and (4) full SCMC, which shows that the complete model consistently performs best, confirming their complementarity.
>
> Due to time limitations, we conducted experimental verification on some datasets. We will complete all experiments in the final version. We hope that the above clarifications and the corresponding text changes will address the reviewer's concerns.
>
> **Revision:**
>
> In **Section 4.3**, we added ablation experiments on MCE and QHC and performed detailed analysis.
> ***
> The above revisions have been marked in blue in the revised version.
>
> If your concerns have been addressed, could you please help raise the score. If you have any other concerns, please let us know, and we will try our best to address them. Thanks.

---

### Official Review · Reviewer_TUhU · 2025-11-04

**Soundness:** 3
**Presentation:** 3
**Contribution:** 2
**Rating:** 4
**Confidence:** 4

**Summary:**

The paper introduces SCMC (Sequence-agnostic Continual Multi-modal Clustering), a framework for continual clustering over streaming modalities that aims to be insensitive to modality arrival order. The method (i) fuses historical and new features via a Residual Fusion Network (RFN), (ii) performs Cross-Temporal Knowledge Collaboration (CTKC) using a matrix cross-entropy regularizer to maximize cross-temporal mutual information, and (iii) applies a Sequence-Agnostic Anti-Forgetting Strategy (SAAS) composed of consistency transfer (CCT) and quality-aware consolidation (QHC) with MMD-based modality weights.

**Strengths:**

1.Clear problem framing: the paper isolates sequence sensitivity as a concrete failure mode for CMC and motivates it with a simple schematic (Fig. 1).
2.FN→CTKC→SAAS is easy to follow; losses and objectives are given explicitly (MCE, CCT, QHC with MMD-based weights). This makes implementation straightforward.

**Weaknesses:**

1.Novelty is incremental and several choices are heuristic. RFN is a standard residual bottleneck; CTKC is essentially an information-uniformity penalty applied across time; SAAS blends L2/KL consistency with contrastive losses and entropy maximization, while “quality” is a monotone transform of MMD to the fused representation. The components are sensible but not particularly original, and the design space (e.g., why MCE vs. CCA/InfoNCE variants, why this MMD→softmax mapping) isn’t justified
2. Sequence-agnostic claim is under-stress-tested. The sequence analysis (Fig. 5) bins permutations into coarse groups; we don’t see exhaustive or adversarial ordering (e.g., worst-first or quality-descending sequences defined by measurable criteria). The t-tests in Fig. 6 compare against two baselines only and focus on p-values rather than effect sizes; there’s no correction for multiple comparisons

**Questions:**

1.Why MCE specifically? Give a principled comparison of CTKC’s MCE objective against alternatives (e.g., InfoNCE between history/new, CCA-style alignment, Barlow Twins-type redundancy reduction). Under what (non-Gaussian) conditions does your Proposition-1/2 reasoning still hold?
2.Since the method claims redundancy suppression, evaluate under heavy noise injection or partial-modality dropout at different steps, and report whether sequence-agnostic gains persist.

**Details Of Ethics Concerns:**

As mentioned above.

---

> ### Author Response · Authors · 2025-11-20
> **Response to Reviewer TUhU (Part I)**
>
> We sincerely appreciate the reviewer’s constructive comments and suggestions.
> ***
> **Weakness 1 and Question 1**
> \
> \
> **(1) The components are sensible but not particularly original.**
>
> The main innovation of this work is a reliable continual multi-modal clustering framework that is insensitive to modality arrival order while preserving trustworthy historical information instead of forgetting it.
>
> Compared with existing continual multi-modal methods:
>
> 1. Unreliable fusion of historical and new information.
> Most approaches simply concatenate or linearly fuse historical and newly arrived modalities, implicitly assuming cross-modal homogeneity and ignoring noise and redundancy. As a result, the shared representation is gradually contaminated and clustering quality degrades over time.
>
> 2. High sensitivity to modality arrival sequence.
> Current methods usually treat all modalities equally during distillation or replay, preserving low-quality signals together with informative ones. This is particularly harmful when high-quality modalities appear early, making the learned clusters highly sensitive to the order in which modalities arrive.
>
> To overcome these limitations, we design:
>
> 1. A reliable continual information propagation module that couples residual fusion with an MCE-based cross-temporal objective, which maximizes mutual information between historical and newly arrived modalities while explicitly suppressing redundant or conflicting signals.
>
> 2. A sequence-agnostic anti-forgetting strategy that links cross-temporal consistency transfer with a quality-aware historical consolidation mechanism, where MMD-based weights emphasize high-quality modalities, attenuate unreliable ones, and thereby stabilize cluster boundaries across different modality orders.
>
>
> **(2) Why MCE in CTKC instead of InfoNCE / CCA / Barlow Twins.**
>
> The design goal of CTKC is to simultaneously (1) maximize cross-temporal dependence between the fused representation $H_f$ and historical/current modalities $(H_f^{\text{old}}, H_t^{\text{new}})$, and (2) suppress both intra-representation and cross-temporal redundancy at the feature-dimension level. This is crucial in continual multi-modal clustering, where the fusion basis must remain expressive yet robust to temporally accumulated noise and modality order changes.
>
> We adopt MCE because it directly regularizes the cross-covariance matrices. In contrast:
>
> 1. InfoNCE between history/new mainly operates at the instance level: it pulls positives closer than negatives but does not explicitly shape the eigenstructure of cross-covariances, so it cannot guarantee isotropy or balanced usage of feature dimensions. It further relies on negative sampling and memory buffers, which are cumbersome in streaming continual learning. In contrast, MCE dispenses with explicit negatives and uses only mini-batch second-order statistics, making it more stable and lightweight.
>
> 2. CCA-style alignment is closely related to our formulation. Under per-view whitening, the MCE term reduces to maximizing the log-determinant of the cross-covariance, i.e., a strictly increasing function of the canonical correlations. Thus CTKC can be seen as a CCA-like criterion where MCE, via a log-det potential and isotropic prior $\tfrac{1}{d}I_d$.
>
> 3. Barlow Twins-type redundancy reduction also exploits covariance structure but matches cross-correlations to the identity via a $\|C - I\|_2^2$ penalty in a static setting. Our MCE instead uses a scale-invariant log-det loss that severely penalizes collapsed eigen-directions ($\log \lambda \to -\infty$) and is explicitly cross-temporal (fusion $\leftrightarrow$ history/current), simultaneously decorrelating $H_f$ and suppressing redundant cross-temporal components.
>
> **(3) Conditions for Proposition 1-2 to hold in the non-Gaussian case.**
>
> In the non-Gaussian setting, we no longer claim that MCE is exactly equal to mutual information. When features are normalized or whitened and dependence is still largely captured by second-order statistics (e.g., elliptically contoured distributions or deep features after BN), MCE can be interpreted as a second-order proxy for mutual information: it shapes the eigenspectrum of the cross-covariance, roughly encoding dependence strength and filtering out redundant dimensions. Only in extreme cases where higher-order nonlinear effects dominate and covariance becomes uninformative does the meaning of Propositions 1–2 turn from a precise theorem into a heuristic interpretation.
>
>
> **(4) Why the MMD $\rightarrow$ softmax mapping.**
>
> For the MMD $\rightarrow$ softmax mapping, we treat MMD as a distributional quality score measuring how close each modality is to the fused representation. We then map negative MMD values through softmax to obtain non-negative, comparable coefficients that sum to one, are monotone in quality, and yield smooth gradients for backpropagation.

---

> ### Author Response · Authors · 2025-11-20
> **Response to Reviewer TUhU (Part II)**
>
> **Weakness 2 and Question 2**
> \
> \
> **(1) Don't see exhaustive or adversarial ordering.**
>
> In the revision, we substantially strengthen this evaluation by adding both exhaustive and adversarial, quality-based orderings. On Caltech3M, we verify that neither worst-first nor quality-descending orders significantly affect SCMC’s clustering performance, whereas existing methods exhibit strong fluctuations under the same conditions, further confirming the anti-sequence sensitivity of our approach.
>
> **(2) Lack of stress test.**
>
> In the revised version, we add a stress test on Caltech3M. We first estimate the quality of each modality via single-view K-means and rank them by clustering accuracy, then deliberately corrupt the best modality by randomly zeroing 50\% of its feature dimensions for all samples, injecting substantial noise and redundancy into the most informative view. The results show that SCMC still outperforms existing CMC methods under this severe degradation and exhibits the smallest performance variance across modality sequences, whereas the CMC baseline suffers clear drops when the degraded view arrives early and fluctuates much more across different orders.
>
>
> **(3) T-test validation baselines are too few; add effect sizes.**
>
> In the revised version, we added baselines to the t-tests to further highlight the advantages of the proposed method and also added the effect size to more clearly show the differences.
>
> **Revisions:**
>
> 1. In **Section 4.2 (Sequence-agnostic Analysis)**, we added exhaustive or adversarial ordering analysis.
>
> 2. In **Appendix A.8 (Stress test)**, we have added stress testing and analysis for SCMC.
>
> 3. In **Section 4.5 (Statistical significance analysis)**, we have added more significance tests and effect sizes.
> ***
>
> The above revisions have been marked in blue in the revised version.
>
> If your concerns have been addressed, could you please help raise the score. If you have any other concerns, please let us know, and we will try our best to address them. Thanks.

---

> ### Author Response · Authors · 2025-11-27
>
> Dear Reviewer  TUhU,
> \
> \
> Thank you for your time in reviewing this work. I sincerely appreciate your earlier feedback, which guided significant improvements in the revised manuscript.
>
> As the rebuttal addresses all raised concerns (mainly including methodological clarifications, additional experiments of sequence-agnostic analysis and statistical significance analysis), I hope these revisions align with your expectations. If possible, I would be grateful if you could re-evaluate the revised manuscript and consider raising your score to reflect the current version's enhancements.
>
> Thank you again for your support throughout this process.
> \
> \
> Best regards
>
> Authors of Paper 4298

---

### Official Review · Reviewer_LZ8w · 2025-11-04

**Soundness:** 3
**Presentation:** 3
**Contribution:** 2
**Rating:** 4
**Confidence:** 3

**Summary:**

This paper tackles Continual Multi-modal Clustering and argues that existing methods are both (i) unreliable when fusing historical and newly arriving modalities and (ii) sequence-sensitive—early, high-quality modalities get forgotten as more (potentially noisy) modalities arrive. The proposed method, SCMC, has three components: (1) a Residual Fusion Network (RFN) that keeps a stable high-rank historical basis while adding a new modality; (2) Cross-Temporal Knowledge Collaboration, which links the fused representation with both historical and current modality features using a matrix cross-entropy objective; and (3) a Sequence-Agnostic Anti-Forgetting Strategy combining Cross-Temporal Consistency Transfer and Quality-aware Historical Consolidation with MMD-based modality importance weights. Experiments on five datasets show strong improvements over baselines and reduced sensitivity to modality order, supported by ablations and sequence-stability plots.

**Strengths:**

- Clear problem framing and motivation around order sensitivity in streaming modalities; the paper operationalizes this with concrete evaluation of different arrival permutations and statistical tests.
- CTKC’s MCE links the fused representation with both historical and current features and is supported by propositions connecting it to mutual information and redundancy control.
- The experiments are comprehensive with consistent empirical gains. Table 1 and ablations show considerable ACC/NMI/PUR improvements; removing SAAS hurts most, underlining the importance of sequence-agnostic design.

**Weaknesses:**

- The method is a combination of standard continual-learning techniques. Each component is individually familiar; what’s new is mostly the combination. As a result, the contribution risks looking like an over-engineered pipeline rather than a principled advance that falls short of ICLR’s bar.
- I personally feel that "sequence-agnostic" is over-claimed in the paper. In continual learning, catastrophic forgetting is expected when training sequentially; numerous CL methods mitigate—but rarely eliminate—this effect, and older knowledge typically decays more than recent. The paper should either temper the claim or provide stronger evidence of genuine order invariance across settings.

**Questions:**

- It’s unclear whether baselines were tuned comparably under continual multi-modal settings. Please detail search ranges and whether per-method tuning was allowed and conduct significance tests.
- Please report params/FLOPs, wall-clock per step, and memory for each ablation and baseline. The log-det term can be cubic; what is d here and how is numerical stability handled?

---

> ### Author Response · Authors · 2025-11-20
> **Response to Reviewer LZ8w (Part I)**
>
> Thank you for your valuable comments.
> ***
> **Weakness 1**
> \
> \
> The main innovation of this work is a reliable continual multi-modal clustering framework that is insensitive to modality arrival order while preserving trustworthy historical information instead of forgetting it.
>
> Compared with existing continual multi-modal methods:
>
> 1. Unreliable fusion of historical and new information.
> Most approaches simply concatenate or linearly fuse historical and newly arrived modalities, implicitly assuming cross-modal homogeneity and ignoring noise and redundancy. As a result, the shared representation is gradually contaminated and clustering quality degrades over time.
>
> 2. High sensitivity to modality arrival sequence.
> Current methods usually treat all modalities equally during distillation or replay, preserving low-quality signals together with informative ones. This is particularly harmful when high-quality modalities appear early, making the learned clusters highly sensitive to the order in which modalities arrive.
>
> To overcome these limitations, we design:
>
> 1. A reliable continual information propagation module that couples residual fusion with an MCE-based cross-temporal objective, which maximizes mutual information between historical and newly arrived modalities while explicitly suppressing redundant or conflicting signals.
>
> 2. A sequence-agnostic anti-forgetting strategy that links cross-temporal consistency transfer with a quality-aware historical consolidation mechanism, where MMD-based weights emphasize high-quality modalities, attenuate unreliable ones, and thereby stabilize cluster boundaries across different modality orders.
>
> ***
> **Weakness 2**
> \
> \
> Our intention with the term "sequence-agnostic" was not to assert strict theoretical invariance for all possible training histories, but rather to emphasize that our method substantially reduces the performance sensitivity to modality order compared to existing continual multi-modal clustering baselines under the considered settings.
>
> To further evaluate sequence agnostic, we conducted two experiments: a stress test and an adversarial ranking experiment (worst-first and quality-descending sequences).
>
> 1. For the stress test, we used K-means to evaluate the quality of each modality in the Caltech3M dataset, considering high accuracy as high quality. Then, we deliberately corrupted the best modality by randomly setting 50% of its feature dimensions to zero, thus injecting strong noise into the most informative view. Experiments show that even under this challenging setting, SCMC consistently outperforms baselines and exhibits minimal performance fluctuations across different rankings. Baselines show a significant performance drop when degraded views appear earlier and exhibit greater overall volatility.
>
> 2. Adversarial ranking experiment. Using the same method as the stress test, we obtained and ranked the quality of different modalities in the Caltech3M dataset, evaluating the clustering performance of worst-first and quality-descending sequences. Experimental results show that SCMC performs very stably in both rankings, while baselines exhibit significant fluctuations.
>
> **Revisions:**
>
> 1. In **Appendix A.8 (Stress test)**, we have added stress testing and analysis for SCMC.
>
> 2. In **Section 4.2 (Sequence-agnostic Analysis)**, we added adversarial ranking experiment analysis.

---

> ### Author Response · Authors · 2025-11-20
> **Response to Reviewer LZ8w (Part II)**
>
> **Question 1**
> \
> \
> Our baselines fall into four groups: (1) classical clustering, (2) traditional MMC, (3) deep MMC, and (4) CMC. Only group (4) is inherently continual; the others are static methods that cannot handle sequential modality arrival. The selection of baselines followed by (Wang et al.).
>
> Wang et al. Adaptcmvc: Robust adaption to incremental views in continual multi-view clustering. CVPR 2025.
>
> For all CMC methods, we tuned hyperparameters in the same continual setting, using shared search ranges for key parameters (learning rate $(1\times10^{-4},3\times10^{-4},1\times10^{-3},3\times10^{-3})$, batch size $(64,128,256)$) and exploring method-specific coefficients around values recommended in the original papers.
> For static baselines (groups (1)–(3)), we kept their original static multi-modal training protocol and tuned their parameters in that regime, since they cannot be trained in a continual manner. These methods are included mainly to show the performance gap between static and continual multi-modal clustering, rather than as direct continual-learning competitors.
>
> In addition, we report significance tests for deep MMC and CMC methods in the revised manuscript.
>
> **Revisions:**
>
> 1. In **Section 4.1 (State-of-the-art Methods)**, we have provided a clearer overview of which methods are used under CMC settings.
>
> 2. In **Section 4.1 (Implementation Details)**, we have added clear search ranges.
>
> 3. In **Section 4.5 (Statistical significance analysis)**, we have added significance tests.
>
> ***
> **Question 2**
>
> 1. In the revised version, we report params, wall-clock per step, and memory for each ablation and baseline. The results show that SCMC achieves high clustering accuracy without significantly increasing computational or memory costs, demonstrating a good performance-efficiency balance.
>
> 2. The log-det term measures the overall uncertainty of a covariance matrix by summing the logarithms of its eigenvalues, thereby quantifying its deviation from a target matrix at a global scale. Here, $d$ denotes the feature dimension. In the revision, we explicitly substitute the MCE log-det form into the RCIP loss and simplify it, making the role and meaning of $d$ transparent.
>
>
> 3. We ensure numerical stability by (1) adding a small diagonal jitter and operating on a positive-definite matrix,
> which keeps all eigenvalues bounded away from zero, and (2) using a stable Cholesky-based routine instead of a raw determinant. Under this setup, we did not observe singularities, NaNs, or exploding values in any experiment.
>
> **Revisions:**
>
> 1. In **Appendix A.10 (Time and space complexity analysis)**, we have provided a clearer time and space complexity analysis.
>
> 2. In **Section 3.1 (Cross-Temporal Knowledge Collaboration)**, we have provided a clearer explanation of the log-det term and maintaining numerical stability.
> ***
> The above revisions have been marked in blue in the revised version.
>
> If your concerns have been addressed, could you please help raise the score. If you have any other concerns, please let us know, and we will try our best to address them. Thanks.

---

> ### Author Response · Authors · 2025-11-27
>
> Dear Reviewer LZ8w,
> \
> \
> Thank you for your time in reviewing this work. I sincerely appreciate your earlier feedback, which guided significant improvements in the revised manuscript.
>
> As the rebuttal addresses all raised concerns (mainly including methodological clarifications, additional experiments of sequence-agnostic analysis and stress test, time and space complexity analysis), I hope these revisions align with your expectations. If possible, I would be grateful if you could re-evaluate the revised manuscript and consider raising your score to reflect the current version's enhancements.
>
> Thank you again for your support throughout this process.
> \
> Best regards
>
> Authors of Paper 4298

---

### Author Response · Authors · 2025-11-20
**General Response**

We appreciate the constructive feedback from the reviewers. The revised manuscript has been uploaded, with all revisions marked in blue.

---

### Author Response · Authors · 2025-11-26

Dear ICLR 2026 SAC, AC, and Reviewers

We would like to express our gratitude to all the reviewers for their valuable feedback, We have carefully considered all suggestions and updated our submission accordingly.

However, we have not yet received responses from Reviewer LZ8w,  TUhU and vdaj. With only few days remaining for discussion, we kindly request your assistance in reading the responses and revised PDF version. It would be greatly appreciated if you could review our rebuttal, as we are ager to know if we have adequately addressed the questions and concerns.

We believe that constructive and timely communication between reviewers and authors is essential for the benefit of both parties.

Thanks a lot for your hard work and support.

Best regards,

Authors of Paper 4298

---

### Author Response · Authors · 2025-12-02

Dear AC,

Thanks for your time and effort in handling our submission. We propose a reliable continual information propagation framework that achieves modality sequence-agnostic, and validate its effectiveness via extensive experiments.

We are pleased reviewers have recognized the value of our work: clear problem framing (**Reviewer LZ8w**, **TUhU**, **vdaj**), innovation and originality (**j6TL**, **vdaj**), effectiveness with solid empirical results (**LZ8w**, **TUhU**, **j6TL**, **vdaj**), theoretical insightfulness (**LZ8w**, **j6TL**, **vdaj**), rigorous theoretical logic (**j6TL**), and clear method implementation (**TUhU**).
***
We have carefully addressed all concerns and revised the work accordingly. Summary of key revisions are:

**Reviewer LZ8w**:

* Clarified the methodological motivation shows that SCMC is not an overcombination in **Section 1 (Introduction)**.
* In **Appendix A.8 (Stress test)**, add stress testing and analysis for SCMC.
* In **Section 4.2 (Sequence-agnostic Analysis)**, add adversarial ranking experiment analysis.
* In **Section 4.1 (State-of-the-Art Methods)**, provid a clearer overview of which methods are used under CMC settings.
* In **Section 4.1 (Implementation Details)**, add clear search ranges.
* In **Section 4.5 (Statistical significance analysis)**,  add significance tests.
* In **Appendix A.10 (Time and space complexity analysis)**,  provid a clearer time and space complexity analysis.
* In **Section 3.1 (Cross-Temporal Knowledge Collaboration)**, provid a clearer explanation of the log-det term and maintaining numerical stability.

**Reviewer TUhU**:

* Clarified the methodological motivation and its differences from existing continual learning approaches, highlighting SCMC's originality in **Section 1 (Introduction)**.
* Clarified the role of RFN residual fusion in **Section 3.1 (Residual Fusion Network)**.
* Explained why MCE is chosen in CTKC and compared it against InfoNCE, CCA, and Barlow Twins.
* Provided the rationale for the MMD-to-softmax mapping.
* In **Section 4.2 (Sequence-agnostic Analysis)**, add exhaustive or adversarial ordering analysis.
* In **Appendix A.8 (Stress test)**, add stress testing and analysis for SCMC.
* In **Section 4.5 (Statistical significance analysis)**, add more significance tests and effect sizes.

**Reviewer j6TL**:

* In **Section 3 (Problem Formulation)**, add a clear notation.
* In **Appendix A.4 (Summary of the proposed SCMC algorithm)**, add a clear pseudocode.
* In **Section 3 (Overall Framework)**, we clearly labeled the correspondence between the loss and the module in Figure 2.
* In **Section 4 (Datasets)**, add a large-scale visual-linguistic dataset and a real-world streaming dataset.
* In **Appendix A.9 (Scalability analyses)**, add scalability analyses.
* In **Section 4.3**, add an ablation on RFN and pretraining.
* In **Section 4.3**, add ablation experiments on MCE and QHC and performed detailed analysis.

**Reviewer vdaj**:

* We have revised **Section 3.1 (Reliable Continual Information Propagation Framework)** to clearly introduce residual fusion networks.
* In the revised manuscript, we have added **Section 3.1 (Cross-Temporal Knowledge Collaboration)** to clearly explain the relationship between MCE loss and mutual information.
* We have added a discussion on differences between SCMC and existing CMC methods in **Appendix A.10**.
* Unified the abbreviation ``residual fusion network (RFN)'' throughout the paper in **Section 3.1 (Residual Fusion Network)**.
* Provided a complete formula definition of MCE upon its first appearance in **Section 3.1 (Cross-Temporal Knowledge Collaboration)**.
Corrected minor notational issues such as writing the activation function as $\sigma(\cdot)$ instead of ``$\sigma = (\cdot, \cdot)$'' in **Section 3.1**.
* Systematically proofread the manuscript to correct typos and formatting issues.
***
After the above response, we received the following feedback:

* **Reviewer LZ8w & TUhU**: We believe that all concerns were addressed, but no responses from reviewers before November 27.
* **Reviewer j6TL**: Confirmed all concerns were addressed and raised the rating to positive rating of 6.
* **Reviewer vdaj**: Maintained a positive rating of 8.
***
In summary, we believe the revised version more clearly articulates the conceptual contribution of SCMC, substantially strengthens the empirical evidence (especially on sequence-agnostic and scalability), and improves the overall clarity and reproducibility of the work. The proposed SCMC is both original and innovative, and we have carefully and comprehensively addressed all of the reviewers’ concerns. Taken together with the strengthened experiments and clarifications in the revision, we are confident that this work now meets the standards required for ICLR'26.
***
We would like to express our sincere thanks again and hope that the AC will consider accepting our paper.

Best regards

Authors of paper 4298

---

### Meta-Review · Area_Chair_mFaF · 2026-01-04

**Summary:**

The reviewers agree that the paper addresses a meaningful and underexplored problem in continual multi-modal learning and presents a technically sound framework with solid empirical results. However, their concerns center on three main issues. First, the novelty is considered incremental, as the method mainly combines several well-known continual learning and information-theoretic components, making the contribution appear more like an engineered pipeline than a principled methodological advance. Second, the core “sequence-agnostic” claim is not sufficiently supported, since the experimental evidence does not convincingly demonstrate true order invariance under diverse or adversarial modality sequences. Third, important design choices and methodological details lack clear justification and exposition, and the experimental analysis misses robustness, efficiency, and scalability evaluations.

**Reviewer Concerns:**

The rebuttal successfully addressed most of the concerns raised by Reviewers LZ8w, TUhU, j6TL, and vdaj, particularly those related to clarification and missing explanations. Specifically, the authors provided clearer descriptions of the Residual Fusion Network (RFN), the role of Matrix Cross-Entropy (MCE) and its connection to mutual information, and the interaction among different loss components, which directly respond to the methodological clarity issues highlighted by these reviewers. The rebuttal also improved the explanation of how different modules contribute to mitigating sequence sensitivity and clarified several notational and presentation issues, addressing concerns about readability and completeness.
In response to the reviewer’s concern that the proposed method is merely a combination of standard continual learning techniques with familiar components and that the novelty lies mainly in their combination, the authors are encouraged to further reflect on this issue and pursue more innovative, principled advances that align with the standards expected by ICLR.

**Reviewer Scores:**

Reviewer j6TL mainly raised concerns about the clarity of the paper’s presentation, the scale of the experiments, the depth of the ablation studies, and the potential redundancy between the loss functions. After the rebuttal, the reviewer would likely increase their score. Reviewer TUhU is likely to maintain their current scores, because although the authors addressed the reviewers’ concerns, the issue that the components are reasonable but not sufficiently novel may lead them to retain their original opinions. Although the authors provided detailed responses to the issues raised by reviewer LZ8w and revised the manuscript accordingly, the reviewer may still maintain their current score, as they consider the proposed method to be a combination of standard continual learning techniques and not a principled advance that meets the ICLR standard.

---

### Decision · Program_Chairs · 2026-01-26

Reject